# Characterizing control of memory CD8 T cell differentiation by BTB-ZF transcription factor Zbtb20

Nicholas K Preiss[1,*], Yasmin Kamal[2,*], Owen M Wilkins[3,4], Chenyang Li[5,6], Fred W Kolling IV[4], Heidi W Trask[4], Young-Kwang Usherwood[1], Chao Cheng[7,8,9], Hildreth R Frost[3], Edward J Usherwood[1]

**Members of the BTB-ZF transcription factor family regulate the immune system. Our laboratory identified that family member Zbtb20 contributes to the differentiation, recall responses, and metabolism of CD8 T cells. Here, we report a characterization of the transcriptional and epigenetic signatures controlled by Zbtb20 at single-cell resolution during the effector and memory phases of the CD8 T cell response. Without Zbtb20, transcriptional programs associated with memory CD8 T cell formation were up-regulated throughout the CD8 T response. A signature of open chromatin was associated with genes controlling T cell activation, consistent with the known impact on differentiation. In addition, memory CD8 T cells lacking Zbtb20 were characterized by open chromatin regions with overrepresentation of AP-1 transcription factor motifs and elevated RNA- and protein-level expressions of the corresponding AP-1 components. Finally, we describe motifs and genomic annotations from the DNA targets of Zbtb20 in CD8 T cells identified by cleavage under targets and release under nuclease (CUT&RUN). Together, these data establish the transcriptional and epigenetic networks contributing to the control of CD8 T cell responses by Zbtb20.**

## Introduction

Through a coordinated effector response, CD8 T cells mediate control over viral or bacterial infection and tumors. This process involves recognition of antigens by naïve CD8 T cells followed by clonal expansion of antigen-specific CD8 T cells into a large pool of effector cells, which elaborate effector molecules to eliminate the pathogen or transformed cells. Subsequently, the effector CD8 T cell ($T_{EFF}$) pool contracts leaving a long-lived memory CD8 T cell ($T_M$)

pool. Memory CD8 T cells are capable of rapidly mounting robust secondary responses and are vital for protection from previously encountered pathogens (Jameson & Masopust, 2009). Because of the protective capacity of the $T_M$, factors controlling their differentiation from naïve to long-lived memory are of great interest for vaccine development and immunotherapy (Butler et al, 2011). Herein, we characterize the contributions of one such factor, the transcription factor Zbtb20 (broad complex, tramtrack, bric-a-brac/poxvirus and zinc finger 20), to the transcriptional and epigenetic control of CD8 T cell differentiation.

Effector and memory CD8 T cell populations are each composed of multiple subsets. During the effector response, CD8 T cells are partitioned along a differentiation spectrum ranging from more terminally differentiated KLRG1$^{hi}$ CD127$^{lo}$ terminal effector cells (TECs) to less terminal KLRG1$^{lo}$ CD127$^{hi}$ memory precursor effector cells (MPECs). As the effector response transitions to the memory phase, most cells with the TEC phenotype undergo apoptosis, whereas MPEC cells are more likely to survive and form $T_M$ (Obar & Lefrançois, 2010; Kaech & Cui, 2012; Herndler-Brandstetter et al, 2018). Memory CD8 T cell subsets include effector memory CD8 T cells ($T_{EM}$) and central memory CD8 T cells ($T_{CM}$). These subsets are recirculating and found in the blood and the spleen. Compared with $T_{EM}$, $T_{CM}$ have a greater proliferative capacity, are less cytotoxic, and can be distinguished from $T_{EM}$ by elevated expression of lymphoid-homing molecules CCR7 and CD62L (Mueller et al, 2013). At memory, CD8 T cells can be further classified by the expression of fractalkine receptor CX$_3$CR1 (Böttcher et al, 2015; Gerlach et al, 2016; Renkema et al, 2020). The expression of CX$_3$CR1 on $T_M$ defines a population of effector-like $T_M$ with elevated levels of granzyme B expression and enhanced killing of infected cells (Böttcher et al, 2015).

Several transcription factors are known to regulate the differentiation process. Among them, Tbet, Zeb-2, and Id2 promote $T_{EFF}$ and/or TEC generation, whereas Zeb1, Tcf1, Eomesodermin, and Runx3 promote $T_M$ and/or MPEC formation (Milner & Goldrath, 2018;

[1]Microbiology and Immunology Department, Geisel School of Medicine at Dartmouth, Lebanon, NH, USA   [2]Department of Medicine, Brigham and Women's Hospital, Harvard Medical School, Boston, MA, USA   [3]Department of Biomedical Data Science, Geisel School of Medicine, Dartmouth College, Hanover, NH, USA   [4]Genomics and Molecular Biology Shared Resource, Dartmouth Cancer Center, Geisel School of Medicine, Lebanon, NH, USA   [5]Genomic Medicine Department, University of Texas MD Anderson Cancer Center, Houston, TX, USA   [6]Graduate School of Biomedical Sciences, The University of Texas MD Anderson Cancer Center UTHealth Houston, Houston, TX, USA   [7]Department of Medicine, Baylor College of Medicine, Houston, TX, USA   [8]Dan L Duncan Comprehensive Cancer Center, Baylor College of Medicine, Houston, TX, USA   [9]The Institute for Clinical and Translational Research, Baylor College of Medicine, Houston, TX, USA

Correspondence: Edward.J.Usherwood@Dartmouth.EDU
*Nicholas K Preiss and Yasmin Kamal contributed equally to this work

Scott & Omilusik, 2019). In addition, we previously characterized the transcription factor Zbtb20 as a regulator of the CD8 T cell differentiation process. Deletion of Zbtb20 during CD8 T cell differentiation resulted in increased MPEC formation, decreased TEC formation, changes in metabolism, and $T_M$ capable of mounting an enhanced secondary response and anti-tumor response (Sun et al, 2020). There are other important signaling events downstream of T cell activation that culminate in activation of transcriptional regulators of T cell function. One such event begins with ligation of the T cell receptor and costimulation. Subsequent signaling results in the activation of the NFAT and AP-1, which together bind composite DNA motifs and synergistically promote the expression of IL-2 and other effector genes (Papavassiliou & Musti, 2020).

The transcription factor Zbtb20 has an N-terminal BTB protein-interaction domain and five C-terminal zinc-finger DNA-binding domains (Zhang et al, 2001; Mitchelmore et al, 2002). Members of this transcription factor family such as Bcl-6, BAZF, and PLZF are known to play important roles in the immune system and in other cell types, regulating cellular differentiation, oncogenesis, and the maintenance of stem cells (Beaulieu & Sant'Angelo, 2011). Importantly, numerous BTB-ZF proteins are key factors in CD4 and CD8 T cell development and function (Cheng et al, 2021). To date, studies on Zbtb20 in lymphocytes other than CD8 T cells have determined that Zbtb20 is essential for B cell differentiation into plasma cells (Chevrier et al, 2014) and for a population of IL-10 producing regulatory T cells (Krzyzanowska et al, 2022). However, few direct genomic targets of Zbtb20 are described (Xie et al, 2008; Zhang et al, 2012; Liu et al, 2013, 2017; Qu et al, 2016) and no direct targets are described in CD8 T cells, confounding efforts to fully characterize the mechanism(s) through which Zbtb20 controls lymphocytes.

In this report, we extended our characterization of the control Zbtb20 exerts over CD8 T cell differentiation. We employed single-cell RNA sequencing (scRNA-seq) coupled with cellular indexing of transcriptomes and epitopes by sequencing (Stoeckius et al, 2017) and single-cell ATAC sequencing (scATAC-seq) to define differences in the transcriptional and epigenetic landscapes of differentiating, Zbtb20-deficient, CD8 T cells. Furthermore, cleavage under targets and release under nuclease (CUT&RUN) (Skene & Henikoff, 2017) was used to identify direct genomic targets of Zbtb20 in CD8 T cells. Together, these techniques determined that Zbtb20-deficient CD8 T cells have a distinct transcriptional and epigenetic profile compared with WT CD8 T cells at both the effector and memory phases of differentiation. In particular, memory CD8 T cells deficient in Zbtb20 up-regulated components of the AP-1 transcription factor complex and had a characteristic AP-1 epigenetic signature. Furthermore, we found that genomic targets of Zbtb20 in CD8 T cells were associated with the regulation of the CD8 T cell response.

# Results

## Zbtb20 controls transcriptional and epigenetic landscapes of effector and memory CD8 T cells

We previously reported that deletion of Zbtb20 in CD8 T cells responding to listeria results in a greater proportion of MPECs,

enhanced mitochondrial and glycolytic metabolisms, increased mitochondrial fuel flexibility, and larger secondary responses (Sun et al, 2020). Here, we comprehensively profiled the transcriptional changes underlying these observed effects. Naïve Zbtb20 conditional knockout (KO) or WT OT-I cells (WT) were adoptively transferred into recipient mice, which were subsequently infected with LM-ActA-OVA (Fig 1A). At day 10 (effector) and day 30 (memory) post-infection, scRNA-seq and scATAC-seq were performed on KO and WT OT-I cells isolated from recipient spleens and labeled with Total-seq antibodies to CD44, CD62L, CD127, and KLRG1 (Fig 1A). Uniform manifold approximation and projection (UMAP) (McInnes et al, 2020 Preprint) representation and cluster analysis of scRNA-seq and scATAC-seq data identified transcriptomic differences (Fig 1B and D) and epigenetic differences (Fig 1C and E) between the four groups. Transcriptomic differences were more pronounced when comparing days 10 and 30 than when comparing KO and WT at the same timepoint (Fig 1B and D). Epigenetically, WT and KO separated on the basis of differential chromatin accessibility at day 30 and to a lesser extent at day 10 (Fig 1C and E). We observed transcriptomic (Fig 1F and H) and epigenetic (Fig 1G and I) shifts in the distribution of WT and KO that prompted us to further examine timepoint-specific differences between KO and WT.

## Transcriptional profile of effector and memory CD8 T cells is controlled by Zbtb20

Transcriptional differences between KO and WT at day 10 (effector) and day 30 (memory) were further characterized by clustering day 10 KO and WT and day 30 KO and WT independently. At day 10, Total-seq staining for CD127 (Fig 2E and M) and KLRG1 (Fig 2F and M) provided proteomic context for UMAP representations of scRNA-seq data and indicated day 10 clusters 1 and 2 contained most of the TECs and clusters 0 and 3 contained mostly MPECs (Fig 2A and B). Clusters 1 and 2 contained fewer KO than WT cells, whereas cluster 3 contained more KO than WT (Fig 2B). This observation is consistent with our previous report, which indicated KO cells form proportionally fewer TECs and more MPECs (Sun et al, 2020). Comparing total KO and WT cells at day 10 revealed KO cells expressed higher levels of memory-associated transcripts (Ly6a, Cd27, Ccr7, Lef1, Bcl2, Tcf7, Runx3, Eomes, and Il7r) and lower levels of effector T cell-associated transcripts, encompassing multiple members of the killer cell lectin-like receptor family (Klre1, Klrd1, Klrk1, Klrc2, Klrg1, Klrb1c, and Klrc1), effector molecules (Gzma, Gzmb, Prf1, and Ifng), Cx3cr1, and transcription factor Zeb2 (Fig 2I).

Individual clusters at day 10 enriched for WT cells such as clusters 1 and 2 displayed the highest surface levels of KLRG1 as determined by Total-seq staining (Fig 2B, F, and M). Furthermore, clusters 1 and 2 were both distinguished from other clusters by elevated transcripts for Zeb2, Gzmb, Cx3cr1, and S1pr5 (Fig 2K). Cluster 1 was distinct in its increased expression of Id2, whereas cluster 2 had higher expression of effector molecule Gzma (Fig 2K). Clusters 0 and 3 were CD127+ by Total-seq labeling and low for KLRG1 (Fig 2E, F, and M). Clusters 0 and 3 expressed higher transcript levels of Cd27, Fosb, Il7r, and Tcf7 (Fig 2K). Cluster 0 was also characterized by significantly elevated levels of Btf3, Bcl2, and Cxcr3 (Fig 2K).

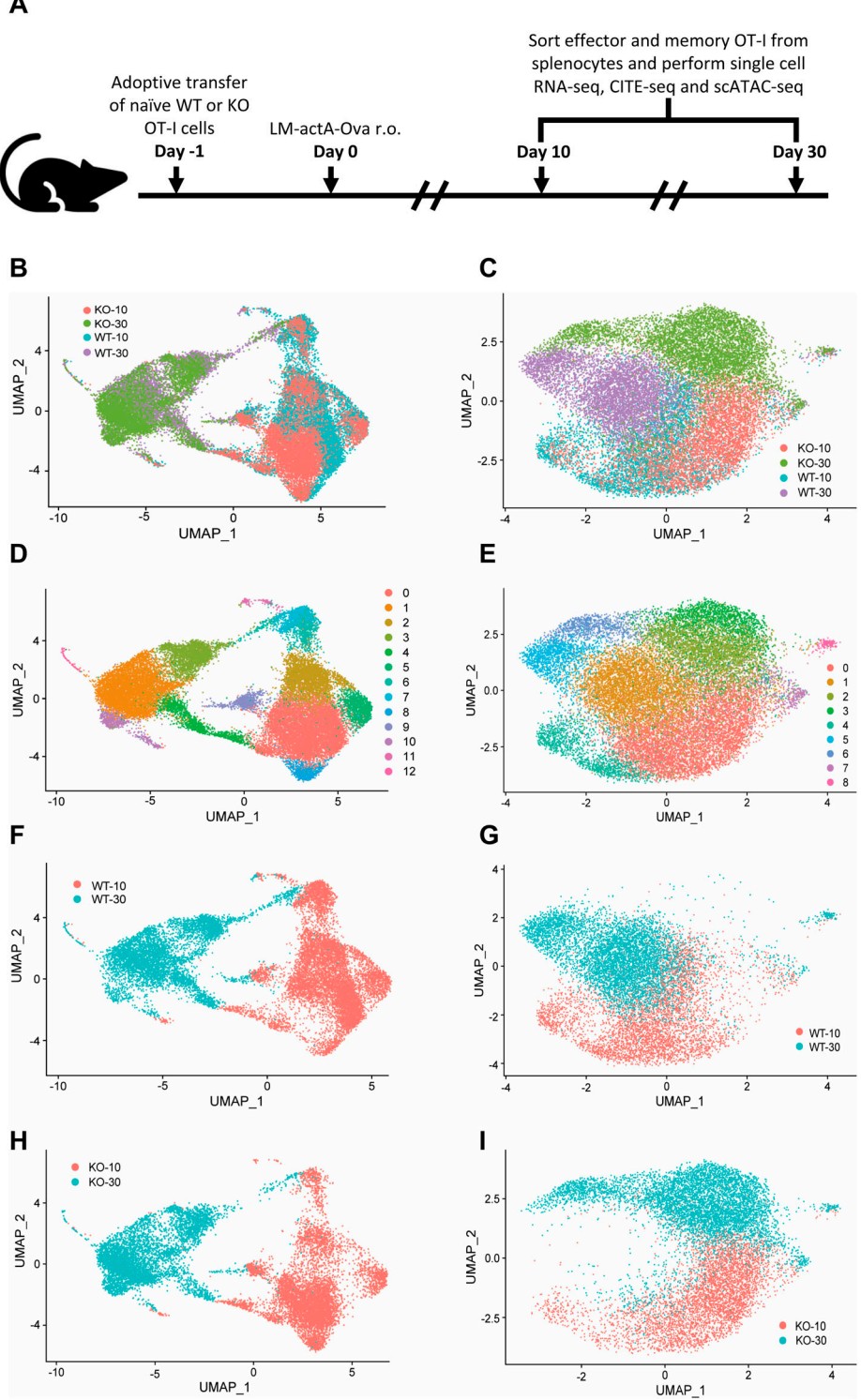

**Figure 1. Transcriptomic and epigenetic differences between effector and memory CD8 T cells with and without Zbtb20.**
**(A)** Mice received naive WT OT-I (WT) or Zbtb20-deficient OT-I cells (KO) and were then infected with LM-actA–OVA. Spleen cells were harvested during the effector (day 10) and memory response (day 30), OT-I cells purified, and single-cell cellular indexing of transcriptomes and epitopes by sequencing/RNA-seq/ATAC-seq performed as described. **(B, C)** KO and WT cells collected from days 10 and 30 represented in the same uniform manifold approximation and projection space by transcript expression (B) and chromatin accessibility (C). **(D, E)** Cluster analysis of KO and WT cells collected from days 10 and 30 in the same uniform manifold approximation and projection space by transcript expression (D) and chromatin accessibility (E). **(F, G)** The distribution of days 10 and 30 WT by transcript expression (F) and chromatin accessibility (G). **(H, I)** The distribution of days 10 and 30 KO by transcript expression (H) and chromatin accessibility (I).

Uniquely identifying transcripts in cluster 3 included *Dapl1*, *Ccr7*, *Fos*, *Jun*, and *Sell* (Fig 2K). We conclude from our analysis of day 10 differences between KO and WT cells that KO cells are characterized by higher expression of memory-associated transcripts, lower expression of effector-associated transcripts, and are less represented in TEC-associated clusters.

At day 30, clusters 2 and 7 contained most of the KLRG1+ cells, whereas clusters 0, 1, and 5 expressed CD127 with little KLRG1

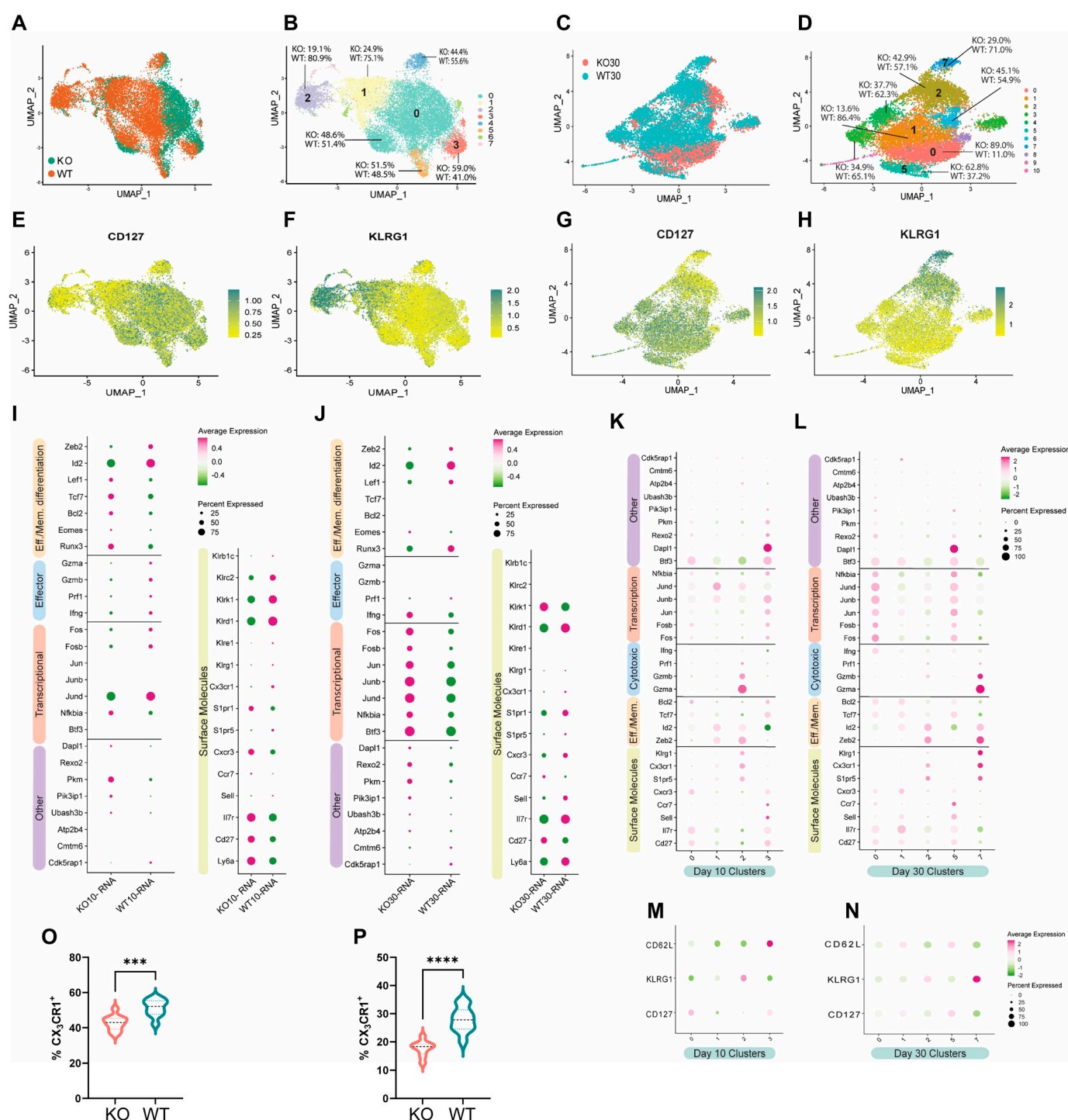

**Figure 2. Transcriptional control of differentiating CD8 T cells by Zbtb20.**
Mice received naive WT OT-I (WT) or Zbtb20-deficient OT-I cells (KO) and were then infected with LM-actA–OVA. Spleen cells were harvested during the effector (day 10) and memory response (day 30), OT-I cells purified, and single-cell cellular indexing of transcriptomes and epitopes by sequencing/RNA-seq performed as described. **(A, C)** Uniform manifold approximation and projection embeddings of merged KO and WT profiles at day 10 (A) and day 30 (C) colored by KO and WT statuses. **(B, D)** Uniform manifold approximation and projection embeddings colored by cluster and displaying distribution of KO and WT cells within each expression cluster at day 10 (B) and day 30 (D). **(E, F, G, H)** Feature plots displaying recovery of antibody derived tags for protein-level expression of surface molecules CD127 and KLRG1 at day 10 (E, F) and day 30 (G, H). **(I, J)** Expression plots comparing gene-level expression of indicated genes between KO and WT at day 10 (I) and day 30 (J) for genes with significant differences in expression. **(K, L)** Expression plots comparing gene-level expression for select clusters at day 10 (K) and day 30 (L). **(M, N)** Expression of antibody-derived tags for CD62L, KLRG1, and CD127 are displayed for select clusters for days 10 (M) and 30 (N). **(O, P)** Flow cytometry was used to detect protein-level expression of CX₃CR1 on KO and WT cells at day 10 and 30 postinfection with LM-actA–OVA. **(O, P)** Percent of KO and WT cells expressing CX₃CR1 at day 10 (O) and day 30 (P). For flow cytometry experiments, results are representative of two independent experiments where n = 4–5 for each condition. *$P \le 0.05$, **$P \le 0.01$, ***$P \le 0.001$, ****$P \le 0.0001$, and ns $P > 0.05$.

expression at both the protein level and the gene level (Fig 2C, D, G, H, L, and N). Most cells in KLRG1-labeled clusters were WT (Fig 2D). Comparison of total KO and WT cells at day 30 indicated that KO cells were strongly enriched for expression of AP-1 components *Fos*, *Fosb*, *Jun*, *Jund*, and *Junb* (Fig 2J). Additional enriched genes in day 30 KO cells include *Btf3* and *Eomes* (Fig 2J). Day 30 WT cells had increased expression of memory genes *Il7r*, *Sell*, and *Ly6a*, but also expressed increased effector genes such as *Zeb2*, *Id2*, *Cx3cr1*, *Klrg1*, *S1pr5*, and *S1pr1* (Fig 2J). Importantly, *Nfkbia* (encoding the repressor of NF-κB, IκBα) was elevated at both day 10 and 30 timepoints in KO cells, consistent with the reported repression of this gene by Zbtb20 (Liu et al, 2013). Strikingly, KLRG1⁻ clusters 0 and 1 contained predominantly KO and WT cells, respectively (Fig 2D). Elevated expression of AP-1 components *Fos*, *Fosb*, *Jun*, *Jund*, and *Junb* formed a strong transcriptional signature distinguishing predominantly KO cluster 0 from predominantly WT cluster 1 (Fig 2L). KO cells were the predominant population in cluster 5 (Fig 2D), a cluster uniquely defined by elevated expression of *Dapl1* (Fig 2L), similar to day 10 cluster 3 (Fig 2K). Cluster 5 shared transcriptional similarities with cluster 0 such as elevated expression of AP-1 components *Fos*, *Fosb*, *Jun*, *Jund*, and *Junb* (Fig 2L). In addition, clusters 5 and 0 shared elevated expression of *Nfkbia*, *Pik3ip1*, and *Rexo2* (Fig 2L). Our analysis at day 30 indicates that Zbtb20-deficient memory CD8 T cells have a signature increase in gene-encoding components of AP-1 and memory-associated genes while preserving reduced expression of key effector-associated transcripts.

Decreased transcript expression of fractalkine receptor *Cx3cr1* was detected in KO cells at both days 10 and 30. Patterns of CX₃CR1 expression are useful for identifying differences in CD8 T cell differentiation, therefore, we characterized the expression of CX₃CR1 on OT-I populations with flow cytometry. Groups of mice received naïve KO and WT OT-I cells and were subsequently administered LM-actA-OVA as previously described. At days 10 and 30 post-infection, splenocytes were collected and the expression of CX₃CR1 on KO and WT OT-I cells was identified using flow cytometry. We observed a decreased percentage of KO cells expressing CX₃CR1 at both day 10 (Fig 2O) and day 30 (Fig 2P). Analysis of the protein-level expression of CX₃CR1 confirmed transcriptomic findings and indicates skewing of the response toward memory in KO cells extends to this important chemokine receptor.

## Compass algorithm predicts metabolic systems regulated by Zbtb20 in CD8 T cells

In our previous work, we identified an association between Zbtb20 and regulation of metabolism in CD8 T cells responding to infection (Sun et al, 2020). Pathway-level analysis of scRNA-seq data utilizing the variance-adjusted Mahalanobis (VAM) method (Frost, 2020) indicated a number of pathways associated with metabolism from both the Molecular Signature Database Gene Ontology (C5) and Hallmark collections were increased at day 10 in KO cells (Figs S1A and B and S2A and B). At day 30, we noted that certain Gene Ontology pathways related to reactive oxygen species, aerobic respiration, and electron transport chain were increased in KO cells (Figs S3A and B), whereas Hallmark pathways for xenobiotic metabolism, adipogenesis, bile acid metabolism, oxidative phosphorylation, and glycolysis were increased in WT cells (Fig S4A and

B). We further used the Compass algorithm to predict differences in the cellular metabolic states between KO and WT cells at days 10 and 30. Compass (Wagner et al, 2021), a flux balance analysis framework (Orth et al, 2010), is optimized for predicting the potential activity of a metabolic reaction from single-cell genomics data. To do this, Compass accounts for metabolic gene expression in scRNA-seq datasets and the global metabolic reaction network from the Recon2 (Thiele et al, 2013) metabolic reconstruction database. We first compared WT cells from the day 10 (effector) and day 30 (memory) timepoints (Fig S5A and B) to establish a Compass metabolic signature for memory versus effector CD8 T cells. Pathways were ranked according to the difference between the percentage of reactions significantly up-regulated and the percentage of reactions significantly down-regulated at day 30 versus day 10 (Fig S5A). The differences in percentages of reactions up-regulated and down-regulated were visualized in a rank plot (Fig S5B). In total, there were 20 pathways with an increased percentage of reactions up at day 30, three pathways with no difference in percentage of reactions up at day 30, and 34 pathways with an increased percentage of reactions up at day 10 (Fig S5A and B). The top five pathways with an increased percentage of elevated reactions at day 30 included vitamin B6 metabolism, biotin metabolism, valine–leucine–isoleucine metabolism, starch and sucrose metabolisms, and histidine metabolism (Fig S5A). The top five pathways with an increased percentage of reactions elevated at day 10 were N-glycan degradation, ROS detoxification, vitamin D metabolism, chondroitin sulfate degradation, and keratan sulfate degradation (Fig S5A). Furthermore, this analysis associated pathways with day 10 WT cells, such as pentose phosphate pathway, glycolysis/gluconeogenesis, and pyruvate metabolism, that are known to be important for effector CD8 T cells (Maciolek et al, 2014; Menk et al, 2018) (Fig S5A). Likewise, pathways important for memory CD8 T cells, such as fatty acid oxidation, fatty acid synthesis, and glutathione metabolism were elevated in day 30 WT cells (O'Sullivan et al, 2014; Ma et al, 2018) (Fig S5A).

Comparing KO and WT at day 10 by Compass, most of the pathways (51 of 57) had increases in the percentage of elevated reactions in KO cells, whereas 4 of 57 pathways had an increased percentage of down-regulated reactions in KO cells (Fig 3A and B). This finding is consistent with our observations from the pathway analysis which identified a number of metabolic pathways up-regulated in KO cells (Fig S1A and B). Top elevated pathways from Compass included propanoate metabolism, N-glycan synthesis, N-glycan degradation, purine synthesis, and CoA synthesis. At day 30, most of the pathways (36 of 57) had a decreased percentage elevated reactions in KO cells, whereas 18 of 57 pathways had an increased percentage of up-regulated reactions in KO cells (Fig 3C and D). The top decreased pathways included ROS detoxification, chondroitin sulfate degradation, heparan sulfate degradation, taurine and hypotaurine metabolisms, and O-glycan synthesis, whereas top increased pathways include cytochrome metabolism, inositol phosphate metabolism, pyrimidine synthesis, folate metabolism, and glyoxylate and dicarboxylate metabolisms (Fig 3C). Interestingly, Compass identified decreases in pathways such as ROS detoxification and oxidative phosphorylation in KO cells (Fig 3C), similar to pathway analysis (Fig S4A and B). However, Compass predicted an increase in pathways related to glycolysis such as pyruvate

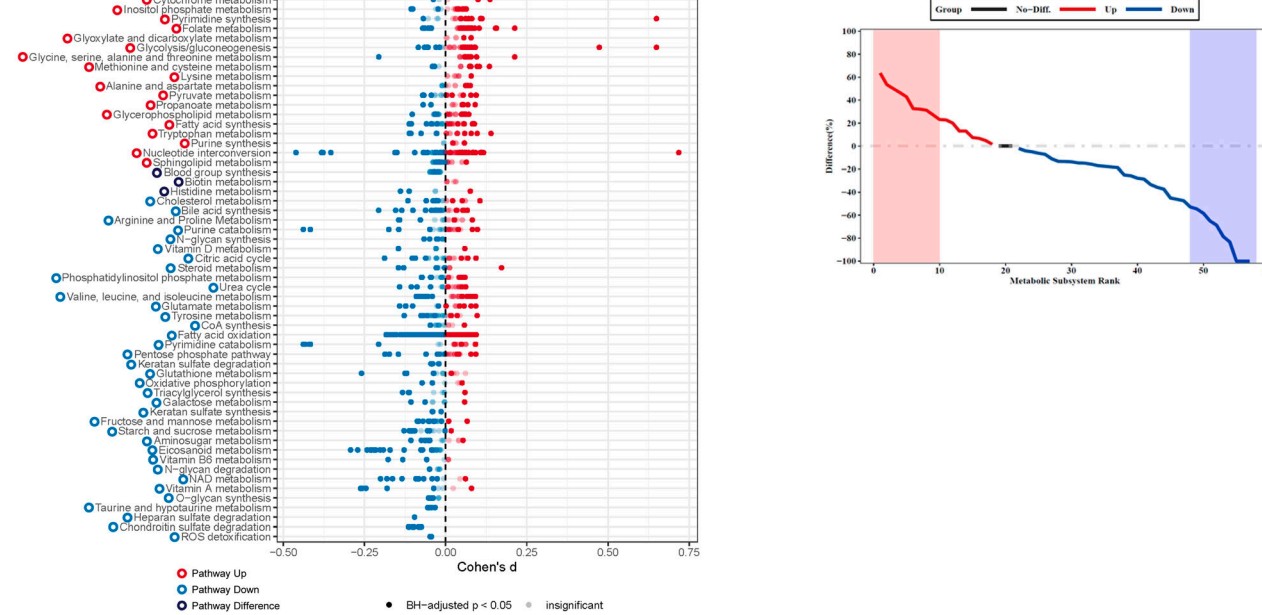

**Figure 3. Zbtb20 controls metabolic transcriptome of CD8 T cells.**
COMPASS was used to analyze the KO and WT single-cell RNA sequencing datasets. **(A, C)** Differential activity of metabolic reactions grouped by Recon2 pathways.
**(A, C)** Reactions are colored according to the sign of their Cohen's d statistic, where a positive sign indicates up-regulation in KO versus WT at day 10 (A) and day 30 (C).
**(B, D)** Rank plots displaying the difference between the percentage of reactions significantly up-regulated and the percentage of reactions significantly down-regulated in Zbtb20 KO OT-I versus WT OT-I at day 10 (B) and day 30 (D).

metabolism and glycolysis/gluconeogenesis in KO cells, whereas pathway analysis-indicated Hallmark glycolysis was decreased in KO cells (Fig S4B). Overall, Compass analysis predicts KO cells'

increase metabolic activity in most of the pathways at day 10, and then decrease of metabolic activity in most of the pathways at day 30 relative to WT cells.

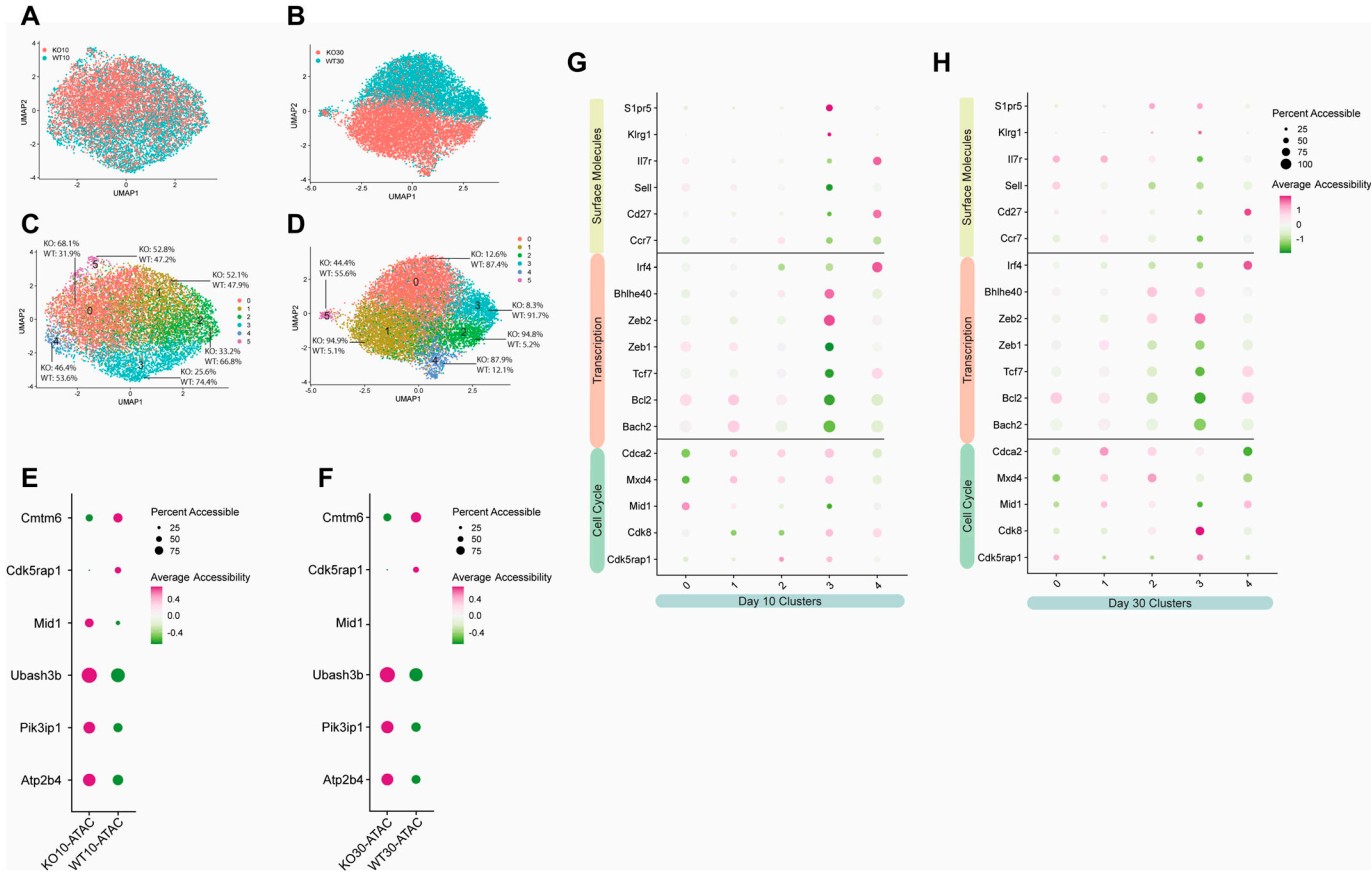

**Figure 4. Epigenetic differences in the absence of Zbtb20.**
Mice received naive WT OT-I (WT) or Zbtb20-deficient OT-I cells (KO) and were then infected with LM-actA–OVA. Spleen cells were harvested during the effector (day 10) and memory response (day 30), OT-I cells purified, and single-cell ATAC sequencing performed as described. **(A, B)** Uniform manifold approximation and projection embeddings of merged KO and WT profiles at day 10 (A) and day 30 (B) colored by KO and WT status. **(C, D)** Uniform manifold approximation and projection embeddings colored by cluster and displaying distribution of KO and WT cells within each expression cluster at day 10 (C) and day 30 (D). **(E, F)** Accessibility plots plots comparing gene-level accessibility of indicated genes between KO and WT at day 10 (E) and day 30 (F). **(G, H)** Accessibility plots comparing gene-level accessibility for select clusters at day 10 (G) and day 30 (H).

## Zbtb20 modulates CD8 T cell epigenetic profile during differentiation

Epigenetic differences between KO and WT at day 10 (effector) and day 30 (memory) were dissected using scATAC-seq by clustering day 10 KO and WT and day 30 KO and WT independently (Fig 4A–D). Comparing total KO and WT at days 10 and 30, we noted significant increases in the accessibility of the WT genome associated with *Cmtm6*, encoding a protein that maintains PD-L1 expression (Burr et al, 2017), and *Cdk5rap1*, encoding a protein that is involved in cell cycle arrest and is associated with the short-lived effector T cell program (Jaeger-Ruckstuhl et al, 2020) (Fig 4E and F). Furthermore, we detected a corresponding increase in transcript expression of *Cdk5rap1* from WT cells at both day 10 and 30 (Fig 2I and J). Comparison of total KO and WT cell genome accessibility indicated an increased accessibility in KO at day 10 of *Mid1*, a gene important for contractile ring formation during cytokinesis (Rincon & Paoletti, 2012), promoting cytotoxic granule exocytosis and CD8 T cell migration (Boding et al, 2014a; 2014b) (Fig 4E). Additional genes with increased accessibility in KO cells at both days 10 and 30 included

*Ubash3b* (also known as suppressor of T cell receptor signaling 1 or Sts1), *Pik3ip1*, and *Atp2b4* (also known as PMCA4) (Fig 4E and F). Broadly, these three genes encode proteins involved in attenuating the CD8 T cell response to antigens. Specifically, *Ubash3b* negatively regulates T cell receptor signaling via dephosphorylation of ZAP70 (Carpino et al, 2004; Wang et al, 2020), *Pik3ip1* restricts T cell activation by inhibiting PI3K/Akt (Uche et al, 2018) and *Atp2b4* is a plasma membrane calcium exporter whose expression attenuates IL-2 production by establishing homeostatic intracellular calcium levels after T cell receptor signaling (Supper et al, 2016). Importantly, the increased chromatin accessibility of these three genes in KO cells corresponded to increased RNA transcription at memory (Fig 2J). Comparison of the epigenetic landscape between total KO and WT cells suggests KO cells have an epigenetic signature associated with the attenuation of T cell activation, consistent with reduced terminal differentiation and promotion of memory.

At day 10, cluster analysis identified WT-majority cluster 3 (Fig 4C) as defined by increased chromatin accessibility in TEC-associated genes such as *Zeb2*, *S1pr5*, and *Klrg1* and decreased chromatin accessibility in MPEC-associated genes such as *Zeb1*, *Tcf7*, and *Bcl2* (Fig 4G). This finding,

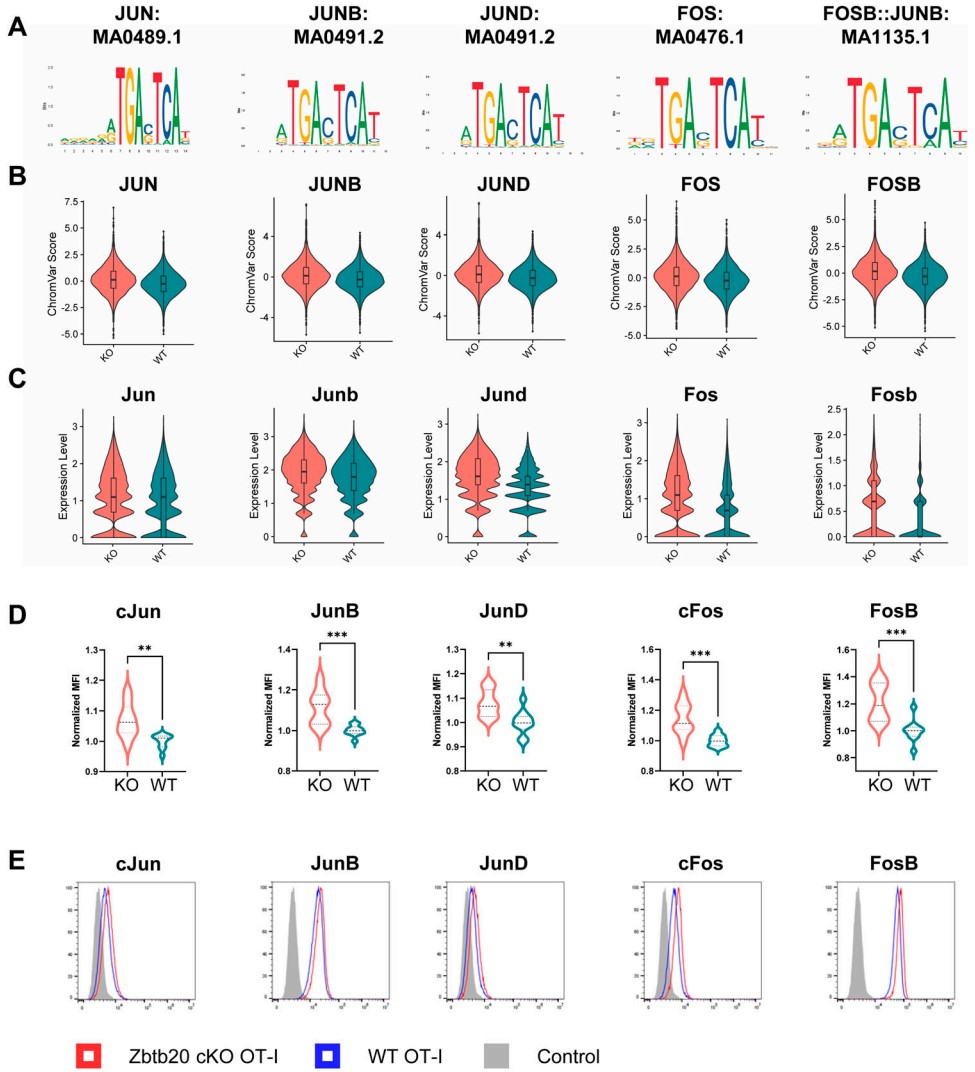

**Figure 5. Zbtb20 controls AP-1 signature in memory CD8 T cells.**
Regions of differentially accessible chromatin between KO and WT were subjected to chromVAR analysis. **(A)** Transcription factor motifs identified by chromVAR in regions of chromatin more accessible in KO cells 30 d postinfection with LM-actA–Ova. **(B)** ChromVAR scores for motifs depicted in (A). **(C)** Violin plots for the RNA expression level of AP-1 transcription factor components at day 30 corresponding to motifs identified by chromVAR analysis. **(D)** Flow cytometry was used to detect protein level expression of AP-1 transcription factor components corresponding to motifs identified by chromVAR analysis 30 d postinfection with LM-actA–Ova. Data were normalized to the mean WT mean fluorescence intensity. **(E)** Representative histograms depicting protein level expression of AP-1 transcription factor components for data presented in (D) compared with control sample stained only with secondary antibody. For flow cytometry experiments, results are shown for two pooled independent experiments where n = 9–10 for each condition. **P ≤ 0.01 and ***P ≤ 0.001.

that TEC-like cluster 3 is poorly populated by KO cells, is consistent with our findings from day 10 transcriptomics, which indicated KO cells were a minority in TEC scRNA clusters (Fig 2). At day 30, KO and WT cells were largely separated into KO clusters 1, 2, and 4 and WT clusters 0 and 3 (Fig 4D). WT cluster 3 and KO cluster 2 were defined by increased accessibility associated with effector genes *Zeb2* and *S1pr5* (Fig 4H), suggesting that these clusters represent effector memory CD8 T cells. However, WT cluster 3 was distinguished from KO cluster 2 by increased accessibility of effector genes *Cdk5rap1* and *Klrg1* and decreased accessibility of memory genes *Bcl2*, *Il7r*, *Zeb1*, *Ccr7*, and *Tcf7* (Fig 4H), indicating that WT cluster 3 is characterized by a more effector-like epigenetic program than KO cluster 2. Thus, WT effector memory CD8 T cells are epigenetically distinct from KO effector memory at day 30.

## Zbtb20 deletion results in AP-1 DNA motif signature in memory CD8 T cells

We next sought to identify transcription factors (TFs) that may be important for mediating the differences we observed in CD8 T cell differentiation in the absence of Zbtb20. To do this, we searched for overrepresented DNA sequence motifs in ATAC peaks associated with KO and WT at days 10 and 30 using chromVAR (Schep et al, 2017); this revealed overrepresentation of pro-memory ZEB1 and TCF7 TF-binding motifs in KO cells at day 10 (Fig S6A and B). We did not detect elevated *Zeb1* transcript expression in KO cells at day 10 (data not shown), but did detect decreased expression of *Zeb2*, which is known to play a reciprocal role to *Zeb1* and mediate terminal effector differentiation (Scott & Omilusik, 2019) (Fig S6C). Furthermore, we did detect elevated transcript expression of the *Tcf7* transcript (Fig S6C). These data suggest increased activity of known memory TFs in KO cells at day 10, consistent with our previous observation that KO forms more MPEC and fewer TEC at day 10.

Overrepresented motifs in KO cells at day 30 included a number of motifs associated with AP-1 TF family members (Fig 5A). Members of the AP-1 TF family regulate CD8 T cell activation, cytotoxicity, and proliferation (Papavassiliou & Musti, 2020). Specifically, motifs associated with JUN, JUNB, JUND, FOS, and

FOSB were detected in areas of chromatin more accessible in KO cells (Fig 5B). Overrepresented AP-1 motifs were generally elevated in KO cells relative to WT cells within individual clusters (Fig S6D). As previously described (Fig 2J), transcripts encoding these AP-1 components are elevated in KO cells (Fig 5C). This prompted us to test the protein-level expression of the identified AP-1 TFs. Antibody staining and flow cytometry on KO and WT cells 30 d after LM-actA-Ova infection revealed that all AP-1 TFs with overrepresented DNA motifs and elevated transcript levels in KO cells had concomitant increases at the protein level (Fig 5D and E). As we had previously detected a decrease in $CX_3CR1$ expression in KO cells, we further tested if AP-1 TFs were differentially expressed between $CX_3CR1^{hi}$ and $CX_3CR1^{lo}$ subsets of memory CD8 T cells. With the notable exception of FosB, AP-1 TF expression was generally elevated in the WT $CX_3CR1^{lo}$ memory CD8 T cell compartment (Fig S7A). This pattern was mirrored in KO memory CD8 T cells (Fig S7B). We also determined that AP-1 TF expression in KO cells was generally elevated in both the $CX_3CR1^{hi}$ and $CX_3CR1^{lo}$ compartments (Fig S7C and D), indicating that elevated AP-1 expression is a general feature of Zbtb20 deletion. Together, these data identify an increase in AP-1 TF activity in Zbtb20-deficient memory CD8 T cells.

### CUT&RUN identifies genomic targets and de novo DNA-binding motifs associated with Zbtb20 in CD8 T cells

The genomic targets of Zbtb20 in CD8 T cells are not known and few targets are described for other cell types (Xie et al, 2008; Zhang et al, 2012; Liu et al, 2013, 2017; Qu et al, 2016). To address this gap in knowledge, we used CUT&RUN (Skene & Henikoff, 2017). This technique generates fragments of DNA associated with transcription factor complexes (expected to be smaller than ~125 bp) and fragments with nucleosome association (expected to be >150 bp) (Skene & Henikoff, 2017). Therefore, DNA fragments from targeting Zbtb20 in CD8 T cells were collected and divided into two size classes, <125 and >150 bp, for analysis (Figs 6A and S8A–E). A similar approach was taken for targeting Zbtb20 in human epithelial kidney (HEK) 293 cells (Fig S9A and B). Combining peaks from both fragment classes, irreproducible discovery rate (IDR) analysis identified 438 peaks in CD8 T cells (Fig S8A) and 688 peaks in HEK 293 cells (Fig S9A). De novo motif analysis of the peak set generated with <125-bp fragments identified two DNA motifs of interest in CD8 T cells (Fig 6B). Interestingly, similar motifs were discovered de novo from analysis of the peak sets generated in HEK 293 cells from <125-bp fragments and >150-bp fragments (Fig S9C–F), however, these motifs were not identified in the CD8 T cell peak set generated from >150-bp fragments. From the CD8 T cell and HEK 293 de novo motif discovery, we deduced core consensus motifs GGAGGCT-GAGGCAGG and GCTGGGA(T/C)TACAGG as associated with Zbtb20 binding.

Next, we used Genomic Regions Enrichment of Annotations Tool (GREAT) (McLean et al, 2010) to analyze the functional significance of cis-regulatory regions identified by Zbtb20-targeted CUT&RUN in CD8 T cells. This analysis highlighted enrichment of Zbtb20 binding events with GO biological process term "negative regulation of immune system process" and GO molecular function term "biotin binding" (Fig 6C). In addition, a number of MGI mouse phenotype

terms associated with abnormal T cell function were detected (Fig 6C). Signal tracks for select genes in GO biological process term "negative regulation of immune system process" (*Lpxn*, *Dusp10*, and *Il10*) and GO molecular function term "biotin binding" (*Pcx* and *Acacb*) were visualized using R package Gviz (Hahne & Ivanek, 2016) (Fig 6D). This analysis indicates Zbtb20-binding events are within cis-regulatory regions functionally associated with immune regulation.

## Discussion

Our previous work examined the transcriptional landscape of Zbtb20-deficient CD8 T cells (KO) compared with WT CD8 T cells during the effector response. Furthermore, we previously demonstrated that Zbtb20 KO memory CD8 T cells mounted a more robust antitumor response (Sun et al, 2020). Here, we extended our analysis of the transcriptional landscape of the CD8 T cell response in the absence of Zbtb20 to the memory response and additionally performed epigenetic analysis at both the effector and memory phases. We also confirmed key observations from our single-cell studies at the protein level and provided the first description, to our knowledge, of Zbtb20 genomic targets in CD8 T cells. This work forms a basis for future studies seeking to harness Zbtb20 in therapeutic applications where the generation of T cell products with increased persistence, a hallmark of memory CD8 T cells, is desirable.

Prior analysis of the effector response identified that the absence of Zbtb20 skewed CD8 T cell differentiation toward the generation of MPECs and away from the generation of KLRG1-expressing TECs (Sun et al, 2020). In this study, we defined broad transcriptional differences between WT and KO CD8 T cells at the memory phase. Similar to KO cells during the effector phase, memory phase KO cells were poorly represented in KLRG1-expressing clusters and predominantly populated clusters expressing CD127. We noted that at both the effector and the memory timepoints, KO cells were the predominant population in clusters 3 and 5, respectively. Both day 10 cluster 3 and day 30 cluster 5 were strongly characterized by the expression of death-associated protein-like 1, *Dapl1*. This is interesting because the expression of *Dapl1* was strongly down-regulated in $T_M$ that formed after secondary stimulation versus $T_M$ that formed after primary stimulation (Wirth et al, 2010). Secondary stimulation of $T_M$ also resulted in more CD62L$^{lo}$ $T_{EM}$ and fewer CD62L$^{hi}$ $T_{CM}$ (Wirth et al, 2010). Thus, *Dapl1* appears to be associated with the $T_{CM}$ program of primary $T_M$. Other studies have shown that Th17 cells with potent tumor clearing capacity and memory CD8 T cells—with both central memory and stem cell memory characteristics—resulting from lymphatic endothelial cell priming highly up-regulating *Dapl1* (Muranski et al, 2011; Vokali et al, 2020). Our data further indicate that *Dapl1* expression in the CD62L$^{hi}$ memory compartment contributes to the identification of a transcriptionally distinct subpopulation of $T_M$-expressing CD62L whose formation is restrained by Zbtb20. Our previous data indicated Zbtb20 KO OT-I primary memory cells had an enhanced recall response to challenge with an orthogonal pathogen expressing ovalbumin peptide (Sun et al, 2020). It is interesting to

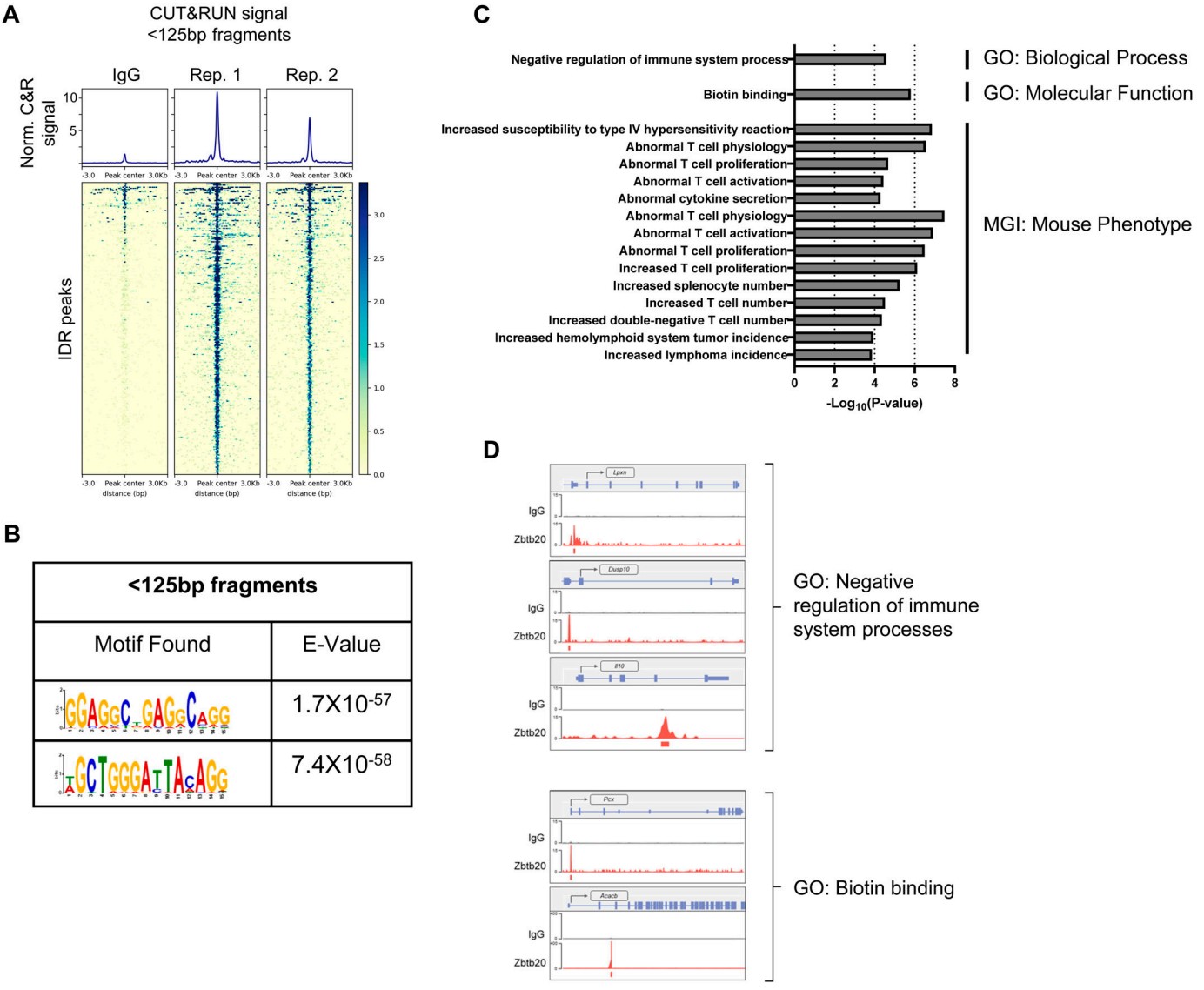

**Figure 6. Cleavage under targets and release under nuclease (CUT&RUN) identifies genomic targets and de novo DNA-binding motifs associated with Zbtb20 binding in CD8 T cells.**
CUT&RUN was used to identify regions of DNA bound by Zbtb20 in CD8 T cells. **(A)** Heat maps and signal of two replicate CUT&RUN datasets for irreproducible discovery rate peaks generated with <125-bp fragments. **(B)** De novo motif discovery analysis in Zbtb20 CUT&RUN peaks in CD8 T cells with E values reported by MEME. **(C)** Enriched terms for ontologies obtained using Genomic Regions Enrichment of Annotations Tool analysis of proximal and distal binding events obtained for Zbtb20 in CD8 T cells. **(D)** Select signal tracks showing Zbtb20 CUT&RUN signal compared with IgG control for genes in GO pathways identified in (C). Peak calls from MACS2 where Zbtb20 CUT&RUN signal was significantly enriched over IgG control are denoted by the peak tracks.

speculate that this enhanced secondary response may be supported by a larger proportion of KO cells differentiating into a *Dapl1*-expressing subset of memory T cells. Future studies will examine the consequences of Zbtb20 deletion on the effectiveness of tertiary and quaternary T cell responses.

Analysis of the transcriptional profiles of KO and WT cells indicated the expression of *Cx3cr1* transcripts, encoding for the fractalkine receptor CX$_3$CR1, was lower in KO cells. This difference was further verified at a protein level by flow cytometry. The work in models of viral infection has noted that either intermediate or high expression of CX$_3$CR1 can be used to classify T$_M$ subsets with differential abilities to self-renew and traffic into peripheral tissues

(Gerlach et al, 2016). In the context of listeria infection, expression of CX$_3$CR1 reportedly identifies a population of T$_M$ that predominantly partitions into the red pulp in the spleen (Renkema et al, 2020), expresses elevated levels of granzyme B, and mediates enhanced target cell killing (Böttcher et al, 2015). The absence of Zbtb20 results in fewer CD8 T cells expressing CX$_3$CR1 during differentiation and may have implications for the effector functions and trafficking potential of the resulting memory population.

A rapid secondary response to previously encountered pathogens is a critical feature of memory CD8 T cells. Epigenetic modifications underly this rapid secondary response. For example, increased epigenetic accessibility is associated with regions of the genome

encoding for effector molecules in memory CD8 T cells relative to naïve cells (Russ et al, 2014; Crompton et al, 2016). In humans, a similar observation has also been made in long-lived, vaccine-induced, yellow fever-specific, memory CD8 T cells (Akondy et al, 2017). In our study, we compared the epigenetic profiles of differentiating CD8 T cells with and without Zbtb20 expression at effector and memory phases. We observed a striking epigenetic signature associated with attenuation of T cell activation in Zbtb20-deficient CD8 T cells. This signature consisted of open chromatin in regions encoding *Ubash3b*, *Pik3ip1*, and *Atp2b4* at both the effector and memory timepoints. At memory, we also noted elevated levels of transcription of these genes in KO cells. It is interesting that this signature consists of genes regulating multiple levels of T cell activation; Ubash3b prevents recruitment of linker for activation of T cells (LAT) by TCR complex-associated ZAP70, Pik3ip1 inhibits PI3K/AKT signaling important for co-stimulation, and Atp2b4 attenuates activation mediated through elevated cytosolic calcium levels (Carpino et al, 2004; Smith-Garvin et al, 2009; Supper et al, 2016; Uche et al, 2018). Downstream of T cell activation through TCR and co-stimulatory signals are transcriptional regulators of the effector response such as ternary NFAT:AP-1 complexes (Macián et al, 2001). In our analysis of motifs overrepresented in areas of chromatin with increased accessibility in memory KO cells, we noted numerous motifs associated with AP-1 binding. Furthermore, RNA- and protein-level expressions of the corresponding AP-1 components were elevated. Interestingly, we observed in our scRNA-seq data that one memory cluster was composed of ~90% WT cells (cluster 1), whereas an adjacent cluster contained ~90% KO (cluster 0) and was defined in part by increased expression of several AP-1 components. This KO cluster 0 appears to be more similar to the previously discussed cluster 5 on the basis of a shared increase in the expression of AP-1 components. The transcription factor activator protein-1 (AP-1) binds DNA as an obligate dimer of two basic leucine zipper (bZIP) transcription factors. The bZIP proteins that form AP-1 dimers include members of the Jun (cJun, JunB, and JunD), Fos (cFos, FosB, Fra-1, and Fra-2), activating transcription factor (ATF2, ATF3, and BATF), and Jun dimerization protein partner (JDP1 and JDP2) families (Shaulian & Karin, 2001). Members of the AP-1 protein family have different abilities to homo or heterodimerize and to transactivate AP-1-responsive genes (Atsaves et al, 2019). The current view is that cellular context and relative expression of AP-1 components determine the complex activity of the AP-1 transcription factor (Eferl & Wagner, 2003). In CD8 T cells, molecular profiling of naïve, effector, memory, and exhausted CD8 T cells identified elevated expression of *Fosb*, *Junb*, and *Fos* in memory CD8 T cells relative to exhausted and effector CD8 T cells (Wherry et al, 2007). Additional work has shown that NFAT orchestrates a program of T cell exhaustion in the absence of AP-1-binding partners (Martinez et al, 2015) through positive regulation of the TOX/NR4A T cell exhaustion axis (Chen et al, 2019; Seo et al, 2019), leading to displacement of AP-1 binding by NR4A (Liu et al, 2019). In the setting of immunotherapy, chimeric antigen receptor (CAR) T cell exhaustion can be ameliorated by adjusting the dosage of particular AP-1 components, namely, cJun in a model of CAR T cell exhaustion (Lynn et al, 2019). Furthermore, AP-1 appears to be critical for maintaining epigenetic priming that supports T cell effector function (Yukawa et al, 2019; Bevington et al, 2020).

Together, these studies highlight the importance of AP-1 component expression for the correct functioning of memory CD8 T cells and CAR T cells. Our data indicate CD8 T cells deficient in Zbtb20 are characterized by increased expression of numerous AP-1 components. The contribution of this elevated AP-1 signature to the enhanced effector cytokine production and secondary responses will be the subject of a future study. An especially pertinent question is how the absence of Zbtb20 affects CD8 T cell exhaustion, as many facets of exhaustion are linked to AP-1. Our AP-1 expression data also indicate that CX$_3$CR1$^{lo}$ memory CD8 T cells are characterized by high expression of AP-1 components cFos, cJun, JunB, and JunD and low expression of FosB. This expression pattern of AP-1 components hints that memory CD8 T cell subsets defined by CX$_3$CR1 may have differential AP-1 component dosage requirements.

The genomic targets of Zbtb20 are poorly characterized with few direct targets described (Xie et al, 2008; Zhang et al, 2012; Liu et al, 2013, 2017; Qu et al, 2016). To our knowledge, no report has examined the chromatin occupancy of Zbtb20 in CD8 T cells. Here, we used the antibody-directed, nuclease-based CUT&RUN procedure (Skene & Henikoff, 2017); a powerful, high-resolution method for determining the chromatin localization of proteins and nucleosome modification patterns, to characterize Zbtb20-binding sites in CD8 T cells and HEK cells (HEK 293). An unbiased motif search in these Zbtb20 CUT&RUN experiments identified two motifs shared between both cell types, indicating these could represent consensus Zbtb20 DNA-binding motifs. Proteins containing BTB protein interaction domains may have multiple protein-binding partners (Maeda, 2016). For example, zinc finger and BTB domain-containing transcription factor BCL6 is known to bind co-repressor proteins through its BTB domain (Huynh & Bardwell, 1998). In the case of Zbtb20, the BTB domain is known to mediate homodimerization, however, binding to other interaction partners remains largely uncharacterized (Mitchelmore et al, 2002). For some transcription factors, binding with different partners is known to change the DNA recognition site (Inukai et al, 2017). Possibly, the two motifs discovered here represent recognition sites for Zbtb20 with different protein-binding partners. Future experiments will seek to experimentally validate Zbtb20 binding to the Zbtb20-associated motifs discovered here and to identify proteins that interact with Zbtb20.

Future studies will further clarify Zbtb20 genomic binding by experimentally testing Zbtb20 binding to motifs discovered in this study. Furthermore, this study indicates the absence of Zbtb20 in CD8 T cells may alter memory recall responses by promoting the formation of a subset of CD62L$^{hi}$ memory cells with high expression of AP-1 components and *Dapl1*. Isolating this subset and comparing its recall capacity with other CD62L$^{hi}$ memory cells will determine the importance of this subset to the control of CD8 T cell fate mediated by Zbtb20.

## Materials and Methods

### Mice and bacteria

Female C57BL/6 mice were purchased from Charles River Laboratories. Zbtb20–fl/fl mice were generated by Dr. WJ Zhang (Second

Military Medical University, China) (Xie et al, 2008). OT-I mice (003831) and CD45.1 mice (002014) were originally purchased from The Jackson Laboratory. Granzyme B (GZB)–cre mice were kindly provided by Dr. R Ahmed (Emory University). CD45.1 OT-I mice and GZB-cre Zbtb20-flox CD45.1 OT-I mice were generated and maintained in-house at Dartmouth College. LM-actA–OVA was kindly provided by Dr. J Harty (University of Iowa). All mice were housed in the Dartmouth Center for Comparative Medicine and Research. The Animal Care and Use Program of Dartmouth College approved all animal experiments.

### Adoptive cell transfer and listeria infection

Splenocytes were harvested from CD45.1 OT-I mice (WT) or GZB-cre Zbtb20-fl/fl CD45.1 OT-I mice (KO) and naive CD8 T cells purified using EasySep Mouse Naive CD8 T Cell Isolation Kits (catalog no. 19858A; STEMCELL Technologies). Fifty thousand naive OT-I cells were retro-orbitally injected into congenic B6 recipient mice. The next day, the recipient mice were retro-orbitally infected with $1 \times 10^6$ CFU LM-actA–OVA.

### Cell preparation for single cell

For isolation of CD8 T cells 10 and 35 d after infection, single-cell suspensions were generated from four mice per recipient group by grinding spleens through nylon filters. CD8 T cells were enriched from these suspensions using Stemcell's EasySep Mouse CD8 T Cell Isolation Kit (#19853). These samples were FcR blocked then stained with antibodies and live/dead stain (LIVE/DEAD™ Fixable Violet Dead Cell Stain Kit, # L34955) for 30 min on ice shielded from light. The antibodies used for cell surface staining from BioLegend were as follows; PE anti-mouse CD8b antibody (YTS156.7.7) and APC anti-mouse CD45.1 antibody (A20). Samples were subsequently washed twice and ~$1 \times 10^6$ congenically marked OT-I cells were purified using fluorescence activated cell sorting for each group of recipients. The samples purified in this way from each group of recipients were then suspended in 100 $\mu l$ buffer and labeled with 1 $\mu g$ per sample of the following Total-seq A antibodies from BioLegend: TotalSeq-A0198 anti-mouse CD127 (A7R34), TotalSeq-A0250 anti-mouse/human KLRG1 (2F1/KLRG1), TotalSeq-A0073 anti-mouse/human CD44 (IM7) and TotalSeq-A0112 anti-mouse CD62L (MEL-14). Samples were labeled for 30 min on ice and subsequently washed with 1 ml PBS twice.

### scRNA-seq

scRNA-seq library preparation was carried out by the Center for Quantitative Biology Single Cell Genomics Core and the Genomics and Molecular Biology Shared Resource at Dartmouth. Droplet-based, 39-end single-cell RNAseq was performed using the 10x Genomics Chromium platform, and libraries were prepared using the Single Cell v3 39 Reagent Kit according to the manufacturer's protocol (10x Genomics). Recovery of Ab-DNA tags (ADTs) from single cells (i.e., cellular indexing of transcriptomes and epitopes by sequencing) was performed by adding 1 ml of ADT additive primer (10 mM, 59-CCTTGGCACCCGAGAATT*C*C-39) to the cDNA amplification reaction and following the 10x protocol for separation of the ADT and mRNA-derived cDNA fractions. ADT libraries were further amplified using 1 ml sample index PCR primer (10 mM, 59- ATGA-TACGGCGACCACCGAGATCTACACTCTTTCCCTACACGACGC*T*C-39) and 1 ml Illumina RPI_X index primer, in which X represents a unique index sequence per sample. ADT and mRNA libraries were normalized to 4 mM and pooled at a 1:9 ratio before loading onto a NextSeq 500 instrument. Libraries were sequenced using 75 cycle kits, with 28 bp on read1 and 56 bp for read2.

### Data analysis for scRNA-seq

The Cell Ranger Single Cell Software Suite (10x Genomics) was used to perform barcode processing and transcript counting after alignment to the mm10 reference genome with default parameters. A total of 7,267 cells in the conditional KO and 10,119 cells in the WT were analyzed for 10,784 genes. Analysis of the gene-level transcript counts' output by Cell Ranger was performed in R (Version 3.5.2 and 4.1.1) on the merged KO and WT datasets (R Core Team, 2018) using the Seurat R package (Version 3.1.4 and 4.1.2) (Butler et al, 2018; Stuart et al, 2019). All ribosomal genes and genes with counts in fewer than 25 cells were excluded. Cells with mitochondrial DNA content >10% or nonzero counts for fewer than 500 genes or more than 3,000 genes were also removed. For day 30, additional contaminant cells were removed by filtering the following genes from the analysis c("C1qa," "C1qb," "C1qc," "Fcer1g," "Cd74"). The filtered gene expression data were normalized using the SCTransform method (Hafemeister & Satija, 2019) and subsequent computations were performed on the matrix of corrected counts. Unsupervised clustering was performed using Seurat implementation of shared nearest neighbor modularity optimization with the resolution parameter set to 0.2 (Waltman & van Eck, 2013). For data visualization, single-cell gene expression data were projected onto a reduced dimensional space as computed by the UMAP (McInnes et al, 2020 Preprint) method for the first 30 principal components of the expression data. The VAM (Frost, 2020) method was used to compute cell-specific scores for pathways from MSigDB collections (Version 7.0) (Subramanian et al, 2005; Liberzon et al, 2011, 2015) that were filtered to remove pathways with fewer than five members or more than 200 members. We identified differentially expressed genes and pathways between KO and WT cells using Wilcoxon rank-sum tests applied to either the normalized counts for each gene or the VAM scores for each pathway with *P*-values adjusted using the Bonferroni method.

### Cellular metabolic analysis using COMPASS

COMPASS is a computational approach to studying cellular metabolic states based on scRNA-seq data and flux balance analysis (Wagner et al, 2021). For this analysis, the CPM (count per million) matrix, filtered as previously described, was passed through COMPASS (v0.9.9.5). The resulting reaction penalty matrix was generated with options "species = mus_musculus, model = RECON2_mat, and-function = mean, penalty-diffusion = knn, $\lambda$ = 0.25, num-neighbors = 30, num-processes = 100, microcluster-size = 30." This resulted in a reaction value of 963 microclusters, which were then assigned to the corresponding cells. The downstream analysis was completed in R (v4.1.0). First, reactions are filtered out that are close to

constant (range < 1 × 10$^{-3}$) or 0 (penalty < 1 × 10$^{-4}$). Then, after the COMPASS documentation, reaction activity scores were obtained by taking the negative log of the output matrix and normalizing by subtracting the minimum value. Next, metabolic differences between Zbtb20 KO OT-I and WT OT-I were determined. For each comparison, unpaired Wilcoxon rank-sum tests and Cohen's D statistic were used to evaluate which reactions are more active in Zbtb20 KO OT-I than in WT OT-I. The resulting *P*-values were adjusted with the Benjamini–Hochberg (BH) method, and reactions with significantly differential activity were defined as having an adjusted *P*-value < 0.05. Subsequently, results of the differential reaction activity scoring were annotated based on RECON2 (Thiele et al, 2013). To examine metabolic differences at the subsystem level, confident reactions with zero or four confidence and subsystems with non-negligible size were visualized with dot plots. Rank plots were generated based on the difference between the percentage of reactions significantly up-regulated and down-regulated in Zbtb20 KO OT-I versus WT OT-I. All diagrams were generated with ggplot2 (v3.3.5).

### scATAC-seq

For scATAC-seq, 1 × 10$^6$ cells were incubated with nuclei lysis buffer for 5′, followed by three washes of 1 ml each in nuclei wash buffer according to 10x Genomics ATAC v1 protocol (CG000169). Nuclei were resuspended in nuclei resuspension buffer (10x Genomics), tagmented, and loaded onto a Chromium Chip E targeting 10,000 cells for capture. Libraries were sequenced on a NextSeq500 instrument targeting 25,000 reads/cell and data processed using the CellRanger-ATAC v1.1 pipeline.

### Data analysis for scATAC-seq

For analysis of scATAC sequencing barcoded fragment files loaded into Signac (Version 4.1.2) (Stuart et al, 2021) for analysis using the standard Signac/Seurat pipeline, fragments were mapped to the peaks and assigned to cells using "Feature Matrix" function. All cell libraries were aligned to the *M. musculus* genome (version mm10). Nucleosome signal strength and transcription start site (TSS) enrichment were determined using "NucleosomeSignal" and "TSSEnrichment." Quality control metric were implemented with removal of cells with nucleosome signal < 4, TSS enrichment > 2, and peak region fragments > 3,000. Macs2 was implemented to call peaks using the Signac "CallPeaks" function per Signac analysis guidelines. Variable features of the peak matrices were used to perform LSI (latent sematic indexing) for dimensionality reduction and subsequent clustering analysis using principal components 2:50. Nonlinear dimensionality reduction and nearest neighbor clustering was implemented using PCs 2:50 for UMAP representation. Gene activity scores were determined by summing fragments intersecting the gene body and promoter region of a gene to infer gene expression from sparser ATAC data. Marker genes for peak-based clustering were generated using Seurat's FindAllMarkers function on gene activity scores. The log-fold change threshold for gene activities was 0.15 when comparing cell types (KO versus WT) and temporal differences (day 10 versus day 30). Log-fold change threshold for peak differences alone was 0.25 mapped to the nearest

gene (50,000 bp up or downstream of the peak). *P*-value threshold for differential gene activity scores and marker genes was set to *P*-value < 0.01.

### ChromVar motif analysis

Signac (Version 4.1.2) was used to test for overrepresentation of DNA motifs in the set of differentially accessible peaks between KO and WT using the ChromVar feature, which allows cell-level motif scoring (Schep et al, 2017). ChromVar z-score deviation scores were plotted and compared between KO and WT cells. We tested for the motifs present in the JASPAR (2020 Version) Motif database (Fornes et al, 2020) for mice.

### CUT&RUN

For CUT&RUN experiments, Zbtb20 with an N-terminal 3XFLAG tag was expressed in HEK 293 cells or primary mouse CD8 T cells by transfection or transduction, respectively, using pCIGAR retroviruses (Huang et al, 2007). CUT&RUN experiments were carried out using EpiCypher's CUT&RUN kit version 1.0 (#14-1048) as directed by the CUTANA user manual version 1.0. Briefly, 500,000 cells (either HEK 293 or CD8 T cells) expressing Zbtb20-3XFLAG were crosslinked with 1% formaldehyde for 5 min and subsequently quenched with 125 mM glycine. Considerations for performing CUT&RUN on crosslinked cells as described in the CUTANA user manual version 1.0 were then applied for the remainder of the procedure. The anti-FLAG (M2) antibody from Sigma-Aldrich was used for detection of Zbtb20-3XFLAG by CUT&RUN.

### Library preparation and sequencing for CUT&RUN

CUT&RUN library preparation and sequencing was performed according to the method described in the CUTANA Library Kit Manual v1.0 (EpiCypher). Briefly, 5 ng of CUT&RUN DNA fragments were prepared for Illumina sequencing using the NEB Ultra II DNA kit with the following modifications: Step 3B.2—1.1x AmPure bead cleanup; Step 4.1.3—PCR amplification with 45 s @ 98°C for initial denaturation followed by 14 cycles of 15 s @ 98°C, 10 s @ 60°C, and 1 m @ 72°C final extension. Libraries were quantified by qubit and run on a Fragment Analyzer (Agilent) for sizing. Libraires were pooled and sequenced to a depth of 10 M, 75 bp paired-end reads/sample on a NextSeq500 instrument.

### CUT&RUN data processing and peak calling

Sequence quality of raw reads was determined using FastQC (v0.11.9) (Andrews, 2010) before read trimming using Cutadapt (v2.4) (Martin, 2011) for adapter sequences with parameters "--nextseq-trim 20 --max-n 0.8 --trim-n -m 1." Bowtie2 (v2.4.2) (Langmead & Salzberg, 2012) was used to map reads to hg38 (for human samples) or mm10 (for mouse samples) with parameters "--local --no-mixed --no-discordant." Unmapped or multi-mapping reads were filtered from alignment files before duplicate identification and removal using MarkDuplicates (Tools, 2022). Samtools (v1.11) (Danecek et al, 2021) was used to sort alignments into nucleosome-free regions (fragment length ≤ 125 bp) or nucleosomal regions (fragment length >

150 bp) and were processed separately in downstream analysis. Peaks calls for ZBTB20 (human samples) and Zbtb20 (mouse samples) were generated using the MACS2 (v2.2.7.1) (Zhang et al, 2008) call peak command in narrowpeak mode with parameters "-f BAMPE --keep-dup all -p 1 × 10$^{-4}$" and -c set to the corresponding IgG IP controls. The *P*-value threshold (-p) was set to the less conservative threshold of 1 × 10$^{-4}$ to facilitate robust identification of reproducible peaks across replicates using the IDR approach, which requires sampling of both signal and noise distributions. IDR (v.2.0.4.2) (Li et al, 2011) was run with parameters "--input-file-type narrowpeak --rank signal.value --peak-merge-method avg." Peaks achieving an IDR value ≤ 0.05 were considered as reproducible and kept for further analysis. Remaining peaks were filtered for those with a fold change at peak summit ≥fourfold to enrich the final peak set for those with the most substantial enrichment above background. Peak calls for H3K4me3 were generated using the same options; however, MACS2 was run in broadpeak mode. For all peak calling, effective genome size was provided using option -g, with a value of 2913022398 for human data, and 2652783500 for mouse. Signal-to-noise was assessed for each sample using fraction of reads in peaks (FRiP), calculated for each sample. Called peaks were filtered for overlap with the ENCODE human blacklist (human: ENCFF356LFX, mouse: ENCFF547MET), mitochondrial regions or unplaced and unlocalized scaffolds. H3K4me3 was used as a positive control in all experiments and produced the expected distribution around TSSs, indicating successful application of the assay. Barcodes counts for each of the CUTANA H3K4 MetStat Spike-in Control dNucs (designer nucleosomes) were determined from FASTQ files to assess on- versus off-target signal. Expected results were obtained in all experiments, indicating that adequate on-target signal was achieved. Spike in *E. coli* DNA was mapped to reference genome K12 MG1655 (obtained from NCBI) using Bowtie2 (v2.4.2) with identical settings to those described for human and mouse. The ratio of unique reads *E. coli*-mapped reads to total reads was used to determine a sample-specific scale factor as described in Orlando et al (2014), and was used for normalization of CUT&RUN signal in downstream analyses. For visualization, normalized signal tracks were generated using deepTools (v 3.3.0) (Ramírez et al, 2016) command BamCoverage with parameters "−binSize 20 −smoothLength 60 −scaleFactor $sf_i$" where $sf_i$ corresponds to the sample-specific scale factor calculated using *E. coli* spike-in DNA. Peak annotation was performed using the *annotatePeak()* function (ChIPseeker [Yu et al, 2015] R-package). Sequences flanking ± 2 kb were used to define promoter regions. R-packages TxDb.Hsapiens.UCSC.hg38.knownGene TxDb.Mmusculus.UCSC.mm10.knownGene were used to annotate peaks to genomic features for human and mouse peaks, respectively. Peaks > 10 kb from the nearest gene were annotated as "distal intergenic." Visualization of specific peak regions and signal intensity was performed in IGV (Robinson et al, 2017).

### De novo motif discovery from peaks

To discover possible enriched sequence motifs among the Zbtb20 CUT&RUN peaks, we performed de novo motif discovery with MEME-ChIP, part of the MEME Suite of analysis tools (Bailey et al, 2015). High-resolution coordinates for peak summits were obtained from peak calling performed on replicate-merged alignment files for nucleosome-free and nucleosomal reads separately. Peaks were called with the MACS2 (v2.2.7.1) call peak command against corresponding IgG samples (option -c) with parameters "-f BAMPE --keep-dup all" and a stringent significance threshold of Q < 0.05 (option -q). Peaks were filtered for those in mitochondrial regions, overlapping with the ENCODE blacklist (see the above section for details) or located on unplaced/unlocalized scaffolds. Sequences flanking ± 250 bp around peak summits were obtained using the subseq() function (from R-package subSeq) in conjunction with R-packages BSgenome.Mmusculus.UCSC.mm10.masked and BSgenome.Hsapiens.UCSC.hg38, for mouse and human peaks, respectively. For peaks located in nucleosome-free regions, MEME-ChIP (v5.4.1) was run with parameters "-order 2 -centrimo-ethresh 0.001 -meme-nmotifs 10 -centrimo-local" and parameter "-ccut 0" to allow for motif discovery across the entire 500-bp region. For peaks located in nucleosomal regions, MEME-ChIP was run with identical parameters except for "-ccut" which was set to 100 to limit motif discovery to the 100-bp surrounding peak summits. To identify motif instances in the reproducible IDR peak set, peaks were scanned for newly discovered motifs using FIMO (Grant et al, 2011) with default options, in addition to parameter "--parse-genomic-coord" and a predetermined background model set with parameter "--bgfile." Background models were determined using command fasta-get-markov with model order parameter "-m" set as 2. Motif matches were considered as those passing a significance threshold of *Q*-value < 0.2.

### GREAT pathways and genes

Peaks from CUT&RUN in CD8 T cells were associated with genes and pathways using GREAT version 4.0.4 (McLean et al, 2010). Association rules were set to basal plus extension with proximal associations set to 5 kb upstream and 1 kb downstream plus distal associations up to 1,000 kb. Curated regulatory domains were included. Pathways found in the Gene Ontology Biological Processes, Gene Ontology Molecular Function, and MGI Mouse Phenotype databases were retrieved.

### Antibody staining

Samples were FcR blocked then stained with antibodies and live/dead stain (LIVE/DEAD Fixable Near-IR Dead Cell Stain, Cat#: L10119; Invitrogen) for 30 min on ice shielded from light. The antibodies used for cell surface staining are as follows; from BioLegend, anti-mouse CD8α BV510 (53-6.7), anti-mouse CD8β PE (YTS156.7.7), anti-mouse CX$_3$CR1 BV421 (SA011F11), anti-mouse CX$_3$CR1 PE (SA011F11), anti-mouse/human KLRG1 FITC (2F1/KLRG1), anti-mouse CD45.1 APC (A20), anti-mouse CD45.1 BV421 (A20), and anti-mouse CD127 APC (A7R34). From Cell Signaling; anti-rabbit IgG (H+L) F(ab')$_2$ fragment AF488 (Cat. #4412), anti-JunB (C37F9), anti-JunD (D17G2), anti-c-Jun (60A8), anti-FosB (5G4), and anti-c-Fos (9F6). After surface staining, all samples were washed twice and either analyzed by flow cytometer or prepared for intracellular staining. Intracellular staining was accomplished by fixing and permeabilizing cells for 30 min on ice using eBioscience fixation/permeabilization

concentrate (Cat#: 00-5123-43; Invitrogen) diluted in fixation/perm diluent (Cat#: 00-5223-56; Invitrogen). Samples were then washed twice with 1X permeabilization buffer (10X Permeabilization Buffer, Cat#: 00-8333-56; Invitrogen), resuspended in 1X permeabilization buffer containing primary antibodies directed against AP-1 Jun and Fos subunits and stained for 30 min on ice. Samples were washed twice with 1X permeabilization buffer then resuspended in 1X permeabilization buffer containing fluorophore-conjugated secondary anti-rabbit IgG and stained for 30 min on ice. Samples were washed twice with 1X permeabilization buffer, resuspended in staining buffer, and analyzed with flow cytometry. All antibodies were used at a dilution of 1:200 except for those used to stain AP-1 components, which were used at 1:100 and the secondary anti-rabbit IgG, which was used at 1:400. Flow cytometry was performed using a CytoFLEX S (Beckman Coulter).

## Statistical analysis

GraphPad Prism 8 was used to analyze all data presented. As designated in the figure legends, either two-tailed paired or unpaired *t* tests were used for comparisons of the two groups. The *P*-values deemed significant are denoted in the figure legends. Numbers of animals used for individual experiments are shown in figure legends. Other statistical analyses were reported by the relevant computational tools.

# Data Availability

All data have been deposited to GEO (accession number GSE234576).

# Supplementary Information

# Acknowledgements

We thank Xiaofeng Wang for helpful insight pertaining to CUT&RUN. CUT&RUN, scRNA-seq, and scATAC-seq were carried out in the Genomics and Molecular Biology Shared Resource (GMBSR) at Dartmouth which is supported by NCI Cancer Center Support Grant 5P30CA023108 and NIH S10 (1S10OD030242) awards. Single-cell studies were conducted through the Dartmouth Center for Quantitative Biology in collaboration with the GMBSR with support from NIGMS (P20GM130454) and NIH S10 (S10OD025235) awards. OM Wilkins is supported by a Cancer Center Core Grant (P30CA023108) from the National Cancer Institute, and the National Institutes of Health-funded Center for Quantitative Biology at Dartmouth (COBRE, 5P20GM130454-03). This work was supported by the BioMT core at Dartmouth: Supported by bioMT through NIH NIGMS grant P20-GM113132. Fluorescence-activated cell-sorting experiments were carried out in DartLab, the Immune Monitoring and Flow Cytometry Shared resource at the Dartmouth Cancer Center, with NCI Cancer Center Support Grant 5P30 CA023108-41. Funding was provided by NIH grants R01 AI122854 (EJ Usherwood), T32 AI007363 (NK Preiss), R21 CA253408, and P20 GM130454 (Y Kamal and HR Frost).

## Author Contributions

NK Preiss: conceptualization, data curation, formal analysis, investigation, visualization, and writing—original draft.
Y Kamal: conceptualization, data curation, formal analysis, investigation, visualization, and writing—review and editing.
OM Wilkins: data curation, formal analysis, visualization, and writing—review and editing.
C Li: data curation, formal analysis, visualization, methodology, and writing—review and editing.
FW Kolling IV: data curation, formal analysis, project administration, and writing—review and editing.
HW Trask: data curation and formal analysis.
Y-K Usherwood: data curation, investigation, and methodology.
C Cheng: data curation, formal analysis, and methodology.
HR Frost: conceptualization, resources, formal analysis, investigation, visualization, methodology, project administration, and writing—review and editing.
EJ Usherwood: conceptualization, supervision, funding acquisition, project administration, and writing—review and editing.

## Conflict of Interest Statement

The authors declare that they have no conflict of interest.

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
