## [Reviewer comments · Life Science Alliance]

Life Science Alliance

Characterizing control of memory CD8 T cell differentiation by BTB-ZF transcription factor Zbtb20

Nicholas Preiss, Yasmin Kamal, Owen Wilkins, Chenyang Li, Fred Kolling IV, Heidi Trask, Young-Kwang Usherwood, Chao Cheng, Hildreth Frost, and Edward Usherwood

DOI: <https://doi.org/10.26508/lsa.202201683>

Corresponding author(s): *Edward Usherwood, Dartmouth College*

Review Timeline:

Submission Date:	2022-08-19
Editorial Decision:	2022-10-19
Revision Received:	2023-03-22
Editorial Decision:	2023-04-20
Revision Received:	2023-06-14
Accepted:	2023-06-15

Transaction Report:

October 19, 2022

Re: Life Science Alliance manuscript #LSA-2022-01683

Prof. Edward John Usherwood
Dartmouth College
Microbiology and Immunology
608E Borwell Building
1 Medical Center Drive
Lebanon, NH 03756

Dear Dr. Usherwood,

Thank you for submitting your manuscript entitled "Characterizing control of memory CD8 T cell differentiation by BTB-ZF transcription factor Zbtb20" to Life Science Alliance. The manuscript was assessed by expert reviewers, whose comments are appended to this letter. We invite you to submit a revised manuscript addressing the Reviewer comments.

Thank you for this interesting contribution to Life Science Alliance. We are looking forward to receiving your revised manuscript.

Sincerely,

Eric Sawey, PhD
Executive Editor
Life Science Alliance
<http://www.lsa-journal.org>

B. MANUSCRIPT ORGANIZATION AND FORMATTING:

Reviewer #1 (Comments to the Authors (Required)):

The manuscript by Preiss et al. reports the transcriptional and epigenetic characterization of Zbtb20-deficient CD8 T cells using CITE-Seq. and scATAC-Seq. In addition, the Zbtb20 binding motifs Lpxn, Dusp10, and IL10 were identified using the CUT&RUN method in the HEK293T cell line and CD8 T cells.

First, the authors analyzed the CITE-Seq data to identify transcriptional differences between wild-type and Zbtb20 knockout CD8 T cells. They discovered that Zbtb20-deficient CD8 T cells retain a memory-prone differentiation phenotype when compared to wild-type CD8 T cells in acute and memory phases of bacterial infection. The gene sets involved in memory T cell differentiation also support the conclusion that the absence of Zbtb20 promotes memory T cell differentiation. Furthermore, the authors showed which metabolic pathways are altered in Zbtb20-deficient CD8 T cells over the course of an infection. They also provided the epigenetic characteristics of CD8 T cells lacking Zbtb20 via scATAC-Seq. The scATAC-Seq data revealed significant differences in the epigenetic landscape of memory signature genes between wild-type and Zbtb20 knockout CD8 T cells. Interestingly, the absence of Zbtb20 increases the level of Cmtm6, which is responsible for the maintenance of surface PD-L1 expression. In addition, they determined that the motifs associated with members of the AP-1 transcription factor family are more accessible in Zbtb20-deficient CD8 T cells. Moreover, the CUT&RUN assay was performed with Zbtb20 to identify its binding motifs in HEK293T cells and CD8 T cells, and it was determined that Zbtb20 binds to the GO in the process of negative regulation of the immune system. Providing the transcriptional and epigenetic landscape of Zbtb20-mediated gene regulation in CD8 T cells, this study will provide valuable resources for future research.

While the composition of the results is advanced following their previous publication, which was published in the Journal of Immunology 2020, the overall manuscript requires further construction and polishing to be accepted by this journal.

Major points

1. In Fig. 2, I would like to suggest that the authors modify the UMAPs to avoid overlapping figures with their previous publication, Journal of Immunology 2020.
2. In Fig. 2I-L, the authors sorted wild-type and Zbtb20 knockout CD8 T cells for the analysis; therefore, when the authors compare the cells, Zbtb20-deficient cells have a greater proportion of memory cells. It appears that the distinct gene profiles are due to the different subset compositions (memory vs effector). I would suggest analyzing the data based on taking advantage of CITE-Seq by looking at the differences based on the Total-Seq markers such as KLRG1 and CD127 so that the authors can compare the MPEC and SLEC (or memory or effector) populations among the identical population.
3. Regarding Fig. 3, I would like to suggest that the authors describe which particular metabolic pathways are involved in memory T cell differentiation or other molecular mechanisms in the published literatures. It would be very informative to the metabolism field in CD8 T cell biology.
4. In Fig. 4E-H, similar to major point 2, I would recommend that the authors compare the epigenetic profiles based on the same clusters so that you can determine which epigenetic changes in the cluster of CD8 T cells are potentially regulated by Zbtb20. Otherwise, it appears to be comparing populations of memory and effector CD8 T cells.
5. In Fig. 5, the authors showed that AP-1 family binding motifs are enriched in Zbtb20 knockout cells; however, it should be analyzed based on the same clusters between the groups to determine whether the AP-1 binding motif enrichment is a result of Zbtb20 deletion. Otherwise, alternative analysis is required to avoid any bias in comparing memory and effector populations.
6. In Fig. 6, the authors showed the binding motifs on the genes associated with the "Negative Regulation of the Immune Process" bind to Zbtb20. It would be more convincing if there were any evidence that Zbtb20-binding genes directly control memory CD8 T cell differentiation. If not, I would suggest conducting an experiment to determine whether overexpression or deletion of the genes results in the same phenotype in CD8 T cell differentiation.

Minor points

1. The illustration of the experimental strategy for CITE-seq and scATAC-seq should be helpful to understand the manuscript better.
2. Why do the same clusters in supplementary Fig. 1A exhibit a different pattern of gene expression between groups?
3. In line 420, the CD44 Total-Seq antibody is omitted.
4. In line 414, the deletion of Zbtb20 does not enhance glycolytic metabolism.
5. In line 435, the percentage of wild-type and knockout cells in cluster 0 is missing from the figure.
6. In line 457, the authors describe Cluster 1 as expressing low levels of KLRG1 and CD127, if so, why does the cluster contain

more wild-type cells?

7. Overall, I would recommend that the authors describe the results in accordance with the order of the figures and be more concise in detail.

8. Does Zbtb20 bind to the genomic regions of the AP-1 transcription family members based on the CUT&RUN assay?

Reviewer #2 (Comments to the Authors (Required)):

Preiss et al characterize the role of transcriptional repressor Zbtb20 in CD8 T cell differentiation by studying immune responses to *Listeria ActA-Ova* in OT-I mice. The authors' previous work reported the finding that Zbtb20 deficiency (in conditional knockout mice) skews the response in favor of memory differentiation (evidenced by higher number of memory precursor cells). Zbtb20 deficiency also seems to cause metabolic changes such as enhanced glycolytic and mitochondrial metabolism. In the current study, the authors extend this observation and present an analysis of the transcriptional and epigenetic landscapes effector and memory CD8 T cells in the presence or absence of Zbtb20. The report relies on single cell genomics (CITE-seq, RNA-seq and ATAC-seq) and mapping the DNA binding sites of Zbtb20 using a recently developed technique of CUT&RUN. Zbtb20 function is as yet poorly characterized in CD8 T cells and the authors current and earlier work addresses this knowledge gap. The current study is descriptive and offers an important resource (for example identifying new pathways to manipulate memory T cell differentiation).

The manuscript is well written and clear as far as the methods and setting up the research question are concerned. The results are mostly descriptive with a list of findings and no definitive conclusions - this is expected given the nature of the analysis. A few comments are listed below:

1. A schematic diagram indicating where the authors think Zbtb20 (and AP-1 TFs) connects with known pathways of T cell activation (downstream of TCR signaling) would help visualize what is stated in the text.
2. Using rapamycin has been shown to enhance the quantity and quality of CD8 T cell memory by Araki et al (Nature, 2009) in mice infected with an acute strain of LCMV. Can the authors comment on their findings in this context? (ie where does Zbtb20 fit in).
3. The authors address a knowledge gap and an interesting fundamental biology question. Can they speculate on biomedical relevance?

Reviewer #3 (Comments to the Authors (Required)):

Preiss et al., in this study investigated the role of ZBTB20 transcription factor on the differentiation of CD8 memory cells using OT-1 and *Listeria* infectious model. The data revealed that in absence of Zbtb20, CD8 favors a transcriptional program linked to memory CD8 cell formation. Using RNA seq, CITE seq analysis and single seq analysis they observed that in zbtb20 deficient CD8TM cells an upregulation of AP1.

1. The major issue of this work is that the data are not fully analyzed and the conclusion for each figure is very simple. We don't know why AP1 is important. They do not investigate as well deeply the role /contribution of Zeb2 or TCF7. Some in vivo experiments might be useful to give more impact and interest of this study (protection, reactivation, epitope stability for example).

With all the data they provide, they should be able to list a precise phenotype of naive vs memory CD8 T cells in absence of Zbtb20 or not. It is really no clear.

2. It is never shown the impact of lack on zbtb20 on CD8TM generation and ratio. This should be shown.

3. The amount of data is really impressive but very hard to read the figures (font too small and extra legends might be essential for the clarity), to understand and to follow in the text. Again most of the time the conclusion of figures rests uneasy.

Sup7 show representative dot plot

missing gating strategy for sorting the cells

a schematic of model would be really useful

Responses to reviewers' comments:

We thank the reviewers for their feedback and we have made changes that we think improve the manuscript. Below we address individually all the points raised by each reviewer. Comments by the reviewers are in black text and our responses are in blue text. Changes to the original texts are marked by yellow highlighting.

Reviewer #1Major points

1. In Fig. 2, I would like to suggest that the authors modify the UMAPs to avoid overlapping figures with their previous publication, Journal of Immunology 2020.

We thank the reviewer for their feedback. The UMAP plots for the Day 10 data have been modified to avoid overlap with our previous publication (see updated Figure 2).

2. In Fig. 2I-L, the authors sorted wild-type and Zbtb20 knockout CD8 T cells for the analysis; therefore, when the authors compare the cells, Zbtb20-deficient cells have a greater proportion of memory cells. It appears that the distinct gene profiles are due to the different subset compositions (memory vs effector). I would suggest analyzing the data based on taking advantage of CITE-Seq by looking at the differences based on the Total-Seq markers such as KLRG1 and CD127 so that the authors can compare the MPEC and SLEC (or memory or effector) populations among the identical population.

In our previous publication (ref. 12), we noted that a major phenotypic difference between Zbtb20 KO and WT CD8 T cells during differentiation was that Zbtb20 KO cells formed fewer KLRG1 positive cells at both effector and memory timepoints. It was unclear, however, the extent to which KO and WT cells were different at an epigenetic and transcriptional level during the course of differentiation at both effector and memory timepoints. In the present manuscript, we clustered both KO and WT cells together in an unbiased manner based on RNA expression. This was done to fully elucidate the transcriptional differences between KO and WT independent of the known differences in KLRG1 expression. A clear example of how this approach can detect heterogeneity within a given subset of cells defined by KLRG1/CD127 expression is captured in our analysis of the day 30 memory timepoint. Here, our analysis indicates that two KLRG1-low clusters - clusters 0 and 1 - are transcriptionally distinct. This is especially important given that the KO/WT ratio in cluster 0 is 89%/11% and 13%/86% in cluster 1. Thus, clustering independent of CITE-seq enabled us to identify a population of KLRG1-low KO cells with a distinct transcriptional profile from KLRG1-low WT cells.

While in principle CITE-seq data for activation/memory markers can be used to identify effector or memory precursor cells, in practice this is problematic. This is due to the difficulty in establishing clear cutoff values for CITE-seq signals that correspond with protein expression defining memory precursor or effector populations. Clustering required using all four CITE-seq markers (KLRG-1, CD127, CD44 and CD62L) and we found the resulting clusters were distinguished by very minor changes in these markers, especially CD44, which we considered were unlikely to be biologically significant (please reference Figure 1 included below the response). It would take an extensive amount of additional experimental work to define biologically relevant cutoffs, which we feel is outside the scope of the current study.

In this paper, we have focused on the transcriptional and epigenetic differences between KO and WT in order to holistically describe the phenotype associated with the CD8 T cell response to infection in the absence of Zbtb20.

Figure 1- Clustering of single cells at day 10 and day 30 based on CITE-seq signal. Clusters, and the distribution of WT and KO cells within these clusters, were identified at both day 10 (A & B) and day 30 (G & H) on the basis of the detection of cell surface molecules by CITE-seq antibodies. Expression of the surface molecules CD127, KLRG1, CD62L and CD44 was visualized at day 10 (C-F) and day 30 (I-L).

3. Regarding Fig. 3, I would like to suggest that the authors describe which particular metabolic pathways are involved in memory T cell differentiation or other molecular mechanisms in the published literatures. It would be very informative to the metabolism field in CD8 T cell biology.

We agree with the reviewer that the application of the Compass algorithm to effector and memory T cells may be a powerful tool for predicting contributions of metabolic pathways to memory CD8 T cell differentiation that are currently unexplored by the field. As part of the effort to validate the use of this tool in CD8 T cells, we provide a short summary of the metabolic pathways that are known to be involved in effector vs memory CD8 T cells in lines 515-520 of the original manuscript. We believe this lends credence to the technique and to other metabolic pathways differentially regulated between effector and memory as well as KO and WT discovered with this analysis.

4. In Fig. 4E-H, similar to major point 2, I would recommend that the authors compare the epigenetic profiles based on the same clusters so that you can determine which epigenetic changes in the cluster of CD8 T cells are potentially regulated by Zbtb20. Otherwise, it appears to be comparing populations of memory and effector CD8 T cells.

Given that single-cell ATAC-seq is not typically run with CITE-seq, and rather a new method, ASAP-seq would be required to truly extract ATAC data in combination with protein detection, we unfortunately will not be able to perform the analysis proposed by the reviewers as the dataset needed to perform to this analysis is not available and only our RNA-seq data has CITE-seq data available.

5. In Fig. 5, the authors showed that AP-1 family binding motifs are enriched in Zbtb20 knockout cells; however, it should be analyzed based on the same clusters between the groups to determine whether the AP-1 binding motif enrichment is a result of Zbtb20 deletion. Otherwise, alternative analysis is required to avoid any bias in comparing memory and effector populations.

Please see the new supplementary figure (Supplemental Fig. 6D) comparing motif enrichment scores of AP-1 motifs across Day 30 clusters between KO and WT. A note has been added to the main text highlighting this finding, lines 593-594. This finding is consistent with what we would expect based on the general increase in transcripts encoding AP-1 components in KO cells. Generally, within a cluster, KO cells seem to be shifted towards the KLRG1-low population, and this population generally seems to express more transcripts for several different AP-1 components. Therefore, finding an increased motif enrichment of AP-1 motifs in KO cells across several different clusters in the scATAC-seq data is consistent with KO cells being shifted across each cluster towards KLRG1-low cells in the scRNA-seq data. Please reference Figure 2 included below:

Figure 2- Gene expression of AP-1 components at day 30. (A & B) Single cells clustered on the basis of gene expression at day 30 and the corresponding overlays of CD127 (IL-7Rα) (C) and

KLRG1 (D). Feature plots were generated for the gene-level expression of Cx3cr1 (F) and several AP-1 components (G-K).

6. In Fig. 6, the authors showed the binding motifs on the genes associated with the "Negative Regulation of the Immune Process" bind to Zbtb20. It would be more convincing if there were any evidence that Zbtb20-binding genes directly control memory CD8 T cell differentiation. If not, I would suggest conducting an experiment to determine whether overexpression or deletion of the genes results in the same phenotype in CD8 T cell differentiation.

Our ATAC-seq data shows overrepresentation of motifs for *Tcf7* and *Zeb1* in Zbtb20 KO CD8 T cells, two transcription factors known to promote memory differentiation. We also detected enrichment of multiple AP-1 family member motifs in KO CD8 T cells. The fact that our cut-and-run experiments did not reveal Zbtb20 binding to genes strongly and directly implicated in memory CD8 T cell differentiation indicates that Zbtb20 likely mediates its effects indirectly. Our ATACseq data implies enhancement of the transcriptional activities of *Tcf7*, *Zeb1* or AP-1 may be responsible. Testing which of these genes are critical for the observed effects of Zbtb20 will be an extensive project, and we consider this to be outside the scope of the current manuscript, which seeks primarily to delineate the transcriptional effects of Zbtb20 deficiency in CD8 T cells.

Minor points

1. The illustration of the experimental strategy for CITE-seq and scATAC-seq should be helpful to understand the manuscript better.

Please see the new Figure 1A with a new experimental schematic.

2. Why do the same clusters in supplementary Fig. 1A exhibit a different pattern of gene expression between groups?

The figure in question pertains to the differential enrichment scores for certain pathways in individual cells constituent to clusters. It does not show expression of individual genes.

3. In line 420, the CD44 Total-Seq antibody is omitted.

This omission has been corrected in the revised manuscript.

4. In line 414, the deletion of Zbtb20 does not enhance glycolytic metabolism.

In our previous publication (ref. 12) we demonstrated that deletion of Zbtb20 enhanced glycolytic metabolism of CD8 T cells responding to listeria in vivo using a Seahorse assay at an effector and memory timepoint. We have added to line 413 of the revised manuscript to clarify that we are referring to the previous publication.

5. In line 435, the percentage of wild-type and knockout cells in cluster 0 is missing from the figure.

In the original manuscript, the percentage KO/WT cells in cluster 0 (day 10 and day 30) is annotated in the figure to the left of cluster 0.

6. In line 457, the authors describe Cluster 1 as expressing low levels of KLRG1 and CD127, if so, why does the cluster contain more wild-type cells?

Our analysis indicates that the KLRG1-low subsets of KO and WT cells form two transcriptionally distinct populations at a memory timepoint. These are represented by cluster 1, which is majority WT, and by cluster 0, which is majority KO. It appears that the transcriptional differences between clusters 1 and 0 are driven in large part by differences in the expression of AP-1 transcription factors. This is described in the original manuscript lines 466-469 and depicted in figure 2L, where clear differences in the gene expression of AP-1 components between clusters 1 and 0 can be visualized.

In summary, we have identified a transcriptionally distinct signature in the KLRG1-low subset of memory cells forming after deletion of *Zbtb20*, such that KO and WT cells with low expression of KLRG1 separate on the basis of transcription into two fairly distinct clusters (1 & 0). It was not possible to make this distinction using only surface staining of the WT and KO memory populations with CD127 and KLRG1.

7. Overall, I would recommend that the authors describe the results in accordance with the order of the figures and be more concise in detail.

Given the multiple assays that were performed using RNAseq, ATACseq and cut-and-run the analyses within and across these platforms are necessarily complex. We have attempted to reduce this as much as possible in the figures to help the reader. However, references to multiple figures in some sections of the text are inevitable to show the concordance between the different approaches taken in this study.

8. Does *Zbtb20* bind to the genomic regions of the AP-1 transcription family members based on the CUT&RUN assay?

We did not observe direct binding of *Zbtb20* to regions associated with AP-1 transcription factors. We speculate that AP-1 is affected downstream of direct *Zbtb20* targets.

Reviewer #2

1. A schematic diagram indicating where the authors think *Zbtb20* (and AP-1 TFs) connects with known pathways of T cell activation (downstream of TCR signaling) would help visualize what is stated in the text.

Our data indicate many important transcriptional and epigenetic changes are induced by the absence of *Zbtb20*. It also appears that a significant proportion of these effects are indirect, rather than mediated directly by *Zbtb20* binding upstream of genes that are key to T cell differentiation. Our data would indicate that *Zbtb20* exerts its effects on T cell differentiation by inducing small changes in multiple key genes, rather than acting predominantly through one or two target genes. The combination of these reasons prohibited us from proposing a model with any level of confidence, as we feel a non-definitive model would not best serve the field.

2. Using rapamycin has been shown to enhance the quantity and quality of CD8 T cell memory by Araki et al (Nature, 2009) in mice infected with an acute strain of LCMV. Can the authors comment on their findings in this context? (ie where does Zbtb20 fit in).

Rapamycin promotes memory CD8 T cell differentiation through inhibition of mTOR, one of the functions of which is to promote metabolic remodeling that favors glycolysis and effector CD8 T cell differentiation. Our Journal of Immunology paper (ref. 12) describing the effect of Zbtb20 deficiency on memory differentiation reported upregulation of the Hallmark mTORC1 signaling and PI3K AKT mTOR signaling pathways, suggestive of Zbtb20 acting upstream of mTOR in influencing metabolic remodeling. Although we observe increased memory differentiation, we also observe increased glycolysis in Zbtb20KO CD8 T cells, possibly suggesting that mTOR stimulation of glycolysis is not sufficient to prevent KO T cells differentiating toward a memory phenotype.

3. The authors address a knowledge gap and an interesting fundamental biology question. Can they speculate on biomedical relevance?

The revised manuscript includes text discussing the biomedical relevance of these findings. These include the fact that understanding how Zbtb20 deficiency promotes memory CD8 T cell differentiation can help us understand how inhibition of Zbtb20 can enhance anti-tumor immunotherapies, supported by our previously published paper in the mouse MC38 tumor model. Lines 651-653 and 658-659.

Reviewer #3

1. The major issue of this work is that the data are not fully analyzed and the conclusion for each figure is very simple. We don't know why AP1 is important. They do not investigate as well deeply the role /contribution of Zeb2 or TCF7. Some in vivo experiments might be useful to give more impact and nterest of this study (protection, reactivation, epitope stability for example).

With all the data they provide, they should be able to list a precise phenotype of naive vs memory CD8 T cells in absence of Zbtb20 or not. It is really no clear.

It is clear from our studies that the transcriptional and epigenetic regulation of CD8 T cells by Zbtb20 is complex, and cannot be explained by direct regulation of a small number of genes that have been shown to be critical for CD8 T cell differentiation. Instead, we delineate the transcriptional pathways that are affected by Zbtb20, to provide a system-wide understanding of the impact of Zbtb20. Studies to test the individual contributions of components of these networks, such as *Tcf7*, *Zeb1* and AP-1 family members will entail extensive further investigation, and we consider those to be outside the scope of the current manuscript.

2. It is never shown the impact of lack on zbtb20 on CD8TM generation and ratio. This should be shown.

These data were shown in our previous paper (ref. 12), summarized in the introduction, lines 94-97, where we show the response elicited by Zbtb20 deficient CD8 T cells results in an

increased proportion of memory precursor cells and a decreased proportion of terminal effector cells.

3. The amount of data is really impressive but very hard to read the figures (font too small and extra legends might be essential for the clarity), to understand and to follow in the text. Again most of the time the conclusion of figures rests uneasy.

The figures in these types of analysis are by nature complex with many different components, necessitating relatively small font sizes in some cases. We have attempted to make the revised figures more easily readable within the confines of the information that is essential to display on each figure.

April 20, 2023

RE: Life Science Alliance Manuscript #LSA-2022-01683R

Prof. Edward John Usherwood
Dartmouth College
Microbiology and Immunology
608E Borwell Building
1 Medical Center Drive
Lebanon, NH 03756

Dear Dr. Usherwood,

Thank you for submitting your revised manuscript entitled "Characterizing control of memory CD8 T cell differentiation by BTB-ZF transcription factor Zbtb20". We would be happy to publish your paper in Life Science Alliance pending final revisions necessary to meet our formatting guidelines.

- please make sure that the author names listed in the manuscript match the author names listed in our system
- please add the author contributions to the main manuscript text
- please consult our manuscript preparation guidelines <https://www.life-science-alliance.org/manuscript-prep> and make sure your manuscript sections are in the correct order
- please use the [10 author names, et al.] format in your references (i.e. limit the author names to the first 10)
- please add a figure legend section to your main manuscript text, including the main and supplementary figure legends
- please add a figure callout for Figure S8 B-E and Figure S9 C-D; please double-check your figure callouts for Figure S1 and Figure S2-you have a callout for panel C, but this panel isn't in the legend or in the figure
- please update your Data Availability statement with the corresponding GEO accession numbers for the uploaded datasets

A. FINAL FILES:

B. MANUSCRIPT ORGANIZATION AND FORMATTING:

Thank you for your attention to these final processing requirements. Please revise and format the manuscript and upload materials within 3 days.

Sincerely,

Reviewer #1 (Comments to the Authors (Required)):

Overall, I understand the viewpoint of the authors, but additional analysis or alternative experiments are required to better comprehend the data. However, each comment explains the authors' scope.

Reviewer #2 (Comments to the Authors (Required)):

In this revised version of the manuscript, the authors have addressed the concerns I raised adequately. I have no further comments.

June 15, 2023

RE: Life Science Alliance Manuscript #LSA-2022-01683RR

Prof. Edward John Usherwood
Dartmouth College
Microbiology and Immunology
Rubin Building Level 7, 710-53
1 Medical Center Drive
Lebanon, NH 03756

Dear Dr. Usherwood,

Thank you for submitting your Research Article entitled "Characterizing control of memory CD8 T cell differentiation by BTB-ZF transcription factor Zbtb20". It is a pleasure to let you know that your manuscript is now accepted for publication in Life Science Alliance. Congratulations on this interesting work.

DISTRIBUTION OF MATERIALS:

Again, congratulations on a very nice paper. I hope you found the review process to be constructive and are pleased with how the manuscript was handled editorially. We look forward to future exciting submissions from your lab.

Sincerely,
